# Gromov-Wasserstein-like Distances in the Gaussian Mixture Models Space

**Antoine Salmona**
*ENS Paris Saclay, CNRS, Centre Borelli UMR 9010*

**Julie Delon**
*Université de Paris, CNRS, MAP5 UMR 8145 and Institut Universitaire de France*

**Agnès Desolneux**
*ENS Paris Saclay, CNRS, Centre Borelli UMR 9010*

**Reviewed on OpenReview:** *https: // openreview. net/ forum? id= 7t7fJT4Gym*

## Abstract

The Gromov-Wasserstein (GW) distance is frequently used in machine learning to compare distributions across distinct metric spaces. Despite its utility, it remains computationally intensive, especially for large-scale problems. Recently, a novel Wasserstein distance specifically tailored for Gaussian mixture models (written GMMs in the paper for the sake of brevity) and known as $MW_2$ (mixture Wasserstein) has been introduced by several authors. In scenarios where data exhibit clustering, this approach simplifies to a small-scale discrete optimal transport problem, which complexity depends solely on the number of Gaussian components in the GMMs. This paper aims to incorporate invariance properties into $MW_2$. This is done by introducing new Gromov-type distances, designed to be isometry-invariant in Euclidean spaces and applicable for comparing GMMs across different dimensional spaces. Our first contribution is the Mixture Gromov Wasserstein distance ($MGW_2$), which can be viewed as a 'Gromovized' version of $MW_2$. This new distance has a straightforward discrete formulation, making it highly efficient for estimating distances between GMMs in practical applications. To facilitate the derivation of a transport plan between GMMs, we present a second distance, the Embedded Wasserstein distance ($EW_2$). This distance turns out to be closely related to several recent alternatives to Gromov-Wasserstein. We show that $EW_2$ can be adapted to derive a distance as well as optimal transportation plans between GMMs. We demonstrate the efficiency of these newly proposed distances on medium to large-scale problems, including shape matching and hyperspectral image color transfer.

## 1 Introduction

The goal of optimal transport (OT) theory is to design meaningful ways to compare probability distributions. It provides very useful mathematical tools for diverse imaging sciences and machine learning tasks including generative modeling (Arjovsky et al., 2017; Genevay et al., 2018; Tolstikhin et al., 2018), domain adaptation (Courty et al., 2016), image processing (Rabin et al., 2012; 2014), and embedding learning (Courty et al., 2018; Xu et al., 2018). For two probability distributions $\mu$ and $\nu$, respectively on two Polish (i.e complete, separable, metrizable) spaces $\mathcal{X}$ and $\mathcal{Y}$, and given a lower semi-continuous function $c\colon \mathcal{X} \times \mathcal{Y} \to \mathbb{R}_+$ called *cost*, optimal transport in its most classic form aims at solving the following optimization problem,

$$\inf_{\pi \in \Pi(\mu,\nu)} \int_{\mathcal{X} \times \mathcal{Y}} c(x,y) \mathrm{d}\pi(x,y) \,, \tag{1}$$

where $\Pi(\mu, \nu)$ is the set of probability measures on $\mathcal{X} \times \mathcal{Y}$ with marginals $\mu$ and $\nu$. When $\mathcal{Y}$ is equal to $\mathcal{X}$, the choice of cost $c_p(x, y) = d_{\mathcal{X}}(x, y)^p$, with $p \geq 1$ and $d_{\mathcal{X}}$ the metric of the space $\mathcal{X}$, induces a distance between probability distributions with finite $p$-th moments, called the Wasserstein distance $W_p$. In the discrete setting, Problem (1) becomes

$$\inf_{\omega \in \Pi(a,b)} \sum_{k,l} C_{k,l} \omega_{k,l} \, ,$$

where $a = (a_1, \ldots, a_m)^T$ and $b = (b_1, \ldots, b_n)^T$ are respectively in the $\mathbb{R}^m$ and $\mathbb{R}^n$ simplexes $\Delta_m$ and $\Delta_n$,[1] $\Pi(a, b) = \{\omega \in \Delta_{m \times n} : \omega \mathbb{1}_n = a \text{ and } \omega^T \mathbb{1}_m = b\}$ and $C$ is a non-negative matrix of size $m \times n$, called cost matrix.

Optimal transport is known to be computationally challenging. Between discrete distributions, its computation involves solving a linear program that rapidly becomes costly as soon as the number of points is moderately large. Between two sets of $n$ points, its computation complexity is $O(n^3 \log(n))$ (Seguy et al., 2017), which compromises its usability for settings with more than a few tens of thousand of points. To lighten OT computational cost, a large number of works have developed efficient computational tools. In particular, Cuturi (2013) proposes to solve an entropic regularized OT problem using the Sinkhorn-Knopp algorithm (Sinkhorn and Knopp, 1967), reducing the cost of the problem to $O(n^2)$. Over the last past years, a large body of works have focused on speeding up the Sinkhorn-Knopp algorithm, building mostly on diverse low-rank approximations (Solomon et al., 2015; Altschuler et al., 2018; 2019; Forrow et al., 2019; Scetbon and Cuturi, 2020; Scetbon et al., 2021). These approaches have helped to reduce the computational cost of the problem from cubic (for the non-regularized problem) to linear complexity. Another type of commonly used solvers are building on sliced mechanisms (Rabin et al., 2012; Kolouri et al., 2019). These solvers average Wasserstein distances between several one dimensional projections of the high-dimensional distributions, leveraging the fact that the OT problem between one-dimentional distributions can be solved using a simple sorting algorithm. Alternatively, Delon and Desolneux (2020) have proposed an OT distance between *Gaussian mixture models* (GMM), called Mixture Wasserstein (MW), where the admissible couplings $\pi$ are themselves constrained to be GMMs. They demonstrated that this specific continuous OT problem could be equivalently reformulated into a discrete version (which had been also proposed independently by Chen et al. (2018)): for two GMM with respectively $K_0$ and $K_1$ components, solving this formulation boils down to solve a small scale $K_0 \times K_1$ discrete OT problem. This distance can be applied to real data by first fitting GMMs on each distribution, making it particularly suited for scenarios where a clustering structure already exists in the data. The main advantage of this approach is that its computational cost arises almost exclusively from fitting the GMMs to the data, since the complexity of the composite OT problem depends neither on the dimension nor on the number of points, but solely on the number of components in the GMMs. This approach offers a scalable and computationally efficient OT distance, which has been used for instance for texture synthesis (Leclaire et al., 2023), evaluating generative models (Luzi et al., 2023), Gaussian Mixture reduction (Zhang and Chen, 2020) or approximate Bayesian computation (Forbes et al., 2021).

One weakness of the classical optimal transport approach lies in the fact that it implicitly assumes that the spaces $\mathcal{X}$ and $\mathcal{Y}$ are *comparable*, i.e. that there exists a relevant cost function $c \colon \mathcal{X} \times \mathcal{Y} \to \mathbb{R}_+$ to compare them. Yet, this assumption is not always verified. For instance, if $\mathcal{X} = \mathbb{R}^d$ and $\mathcal{Y} = \mathbb{R}^{d'}$ with $d \neq d'$, the definition of a meaningful cost function $c \colon \mathbb{R}^d \times \mathbb{R}^{d'} \to \mathbb{R}_+$ is not straightforward. Furthermore, some applications such as shape matching require having an OT distance that is invariant to a given family of transformations, such as translations or rotations, or more generally to *isometries*[2]. Even if the two distributions involved in these applications do live in the same ground space, it is not straightforward to design a cost function such that the resulting OT distance will be invariant to these families of transformations. To overcome those limitations, several non-convex variants of Problem (1) have been proposed (Cohen and Guibasm, 1999; Pele and Taskar, 2013; Alvarez-Melis et al., 2019; Cai and Lim, 2022). Among these, the Gromov-Wasserstein (GW) (Mémoli, 2011) distance is perhaps the most frequently utilized, recently gaining significant attention for the versatility it provides. Indeed, it only requires modeling topological aspects of the distributions within each domain to compare them without having to specify first a subset of invariances nor to design a relevant cost function between the spaces the distributions lie on. The GW problem between two measures $\mu$ and $\nu$

---

[1] The simplex $\Delta_m$ is the subset of $\mathbb{R}^m$ of $x = (x_1, \ldots, x_m)^T$ such that for all $1 \leq k \leq m$, $x_k \geq 0$, and $\sum_{k=1}^m x_k = 1$.
[2] We say that $\phi \colon \mathcal{X} \to \mathcal{Y}$ is an isometry if for all $(x, x') \in \mathcal{X}^2$, $d_{\mathcal{Y}}(\phi(x), \phi(x')) = d_{\mathcal{X}}(x, x')$.

living respectively on $\mathcal{X}$ and $\mathcal{Y}$ aims at solving

$$\inf_{\pi \in \Pi(\mu,\nu)} \int_{\mathcal{X} \times \mathcal{Y}} \int_{\mathcal{X} \times \mathcal{Y}} |c_{\mathcal{X}}(x,x') - c_{\mathcal{Y}}(y,y')|^p \mathrm{d}\pi(x,y) \mathrm{d}\pi(x',y'),$$

where $c_{\mathcal{X}} : \mathcal{X} \times \mathcal{X} \to \mathbb{R}$ and $c_{\mathcal{Y}} : \mathcal{Y} \times \mathcal{Y} \to \mathbb{R}$ are two cost functions. Since this optimization problem only requires to define cost functions in each respective space, it remains very versatile and can be defined with very little assumptions on the spaces $\mathcal{X}$ and $\mathcal{Y}$. This approach has been applied to shape matching (Mémoli, 2009), or more generally to correspondence problems (Solomon et al., 2016), word embedding (Alvarez-Melis and Jaakkola, 2018), graph classification (Vayer et al., 2019a), graph prediction (Brogat-Motte et al., 2022), and generative modeling (Bunne et al., 2019).

Computationally speaking, the Gromov-Wasserstein problem is known to be much more costly to solve than the classic linear OT problem. Indeed, the problem is non convex, quadratic with respect to $\pi$ and known to be NP-hard. One possible approach to solve GW consists in linearizing the cost and to solve iteratively several classic OT problems. Entropic regularization of GW has also been proposed in (Peyré et al., 2016; Solomon et al., 2016) and results in a still non convex problem which can be solved by a projected gradient algorithm, where each projection is itself an entropic linear optimal transport problem. In recent years, several practical approximations of GW have been proposed in the literature to reduce its computational complexity and solve it efficiently, either through quantization of input measures (Chowdhury et al., 2021), recursive clustering approches (Xu et al., 2019; Blumberg et al., 2020), or using a minibatch scheme (Fatras et al., 2021). Specifically to the Euclidean setting, Vayer et al. (2019b) has introduced a solver buiding on a sliced mechanism, and leveraging the observation that the GW problem seems most of the time easy to solve between one-dimensional distributions. More recently, Scetbon et al. (2022) have shown that the low-rank approximations used to speed-up the Sinkhorn-Knopp algorithm were particularly suited for the regularized GW problem, resulting in a much more computationally efficient solver. In this work, we propose to build on the ideas of Delon and Desolneux (2020) in order to construct OT distances between GMMs that are invariant to isometries and that stay relevant between GMMs of different dimensions. These distances share similarities with the one defined in (Chowdhury et al., 2021), since they rely on a form of quantization of the original data through the GMM representation. One of these distances is a "Gromovization" of the Mixture Wasserstein distance, that we call MGW. We will see that the structured representation of MGW makes it very robust in practice, and permits to design an efficient and scalable solver using a fixed small number of Gaussian components, while keeping competitive precision and running times (when compared to the state-of-the-art methods described above) when the number of points of the underlying data increases.

**Contributions of the paper.** In this paper, we introduce two Gromov-Wasserstein type OT distances between GMMs that are designed to be invariant (at least) to isometries. More precisely, we introduce in Section 3 a natural Gromov version of the distance introduced by Chen et al. (2018) and Delon and Desolneux (2020), that we call MGW for *Mixture Gromov Wasserstein*. This distance can be used for applications which only require to evaluate how far the distributions are from each other, without having to identify correspondences between points. However, this formulation does not directly allow to derive an optimal transportation plan between the points. To design a way to define such a transportation plan, we define in Section 4 another distance that we call EW for *Embedded Wasserstein*. This latter turns out to be closely related to the Gromov-Wasserstein distance and coincides with the OT distance introduced by Alvarez-Melis et al. (2019). We show that EW can be adapted to derive a distance and optimal transportation plans between GMMs and we then define a heuristic transportation plan for MGW by analogy with EW. Finally, in Section 5, we illustrate the pratical use of our distances on medium-to-large scale problems such as shape matching and hyperspectral image color transfer and we compare the performance of our methods with other recent GW based approaches, both on assessing distances between clouds on points and drawing correspondences between points. All the proofs are postponed to the appendix.

## Notation

We define in the following some of the notation that will be used in the paper.

- $\langle x, x' \rangle_d$ stands for the Euclidean inner-product in $\mathbb{R}^d$ between $x$ and $x'$. We will use the notation $\langle x, x' \rangle$ when the dimension is clear and unambiguous.
- $\|x\|_{\mathbb{R}^d}$ stands for the Euclidean norm of $x \in \mathbb{R}^d$. We will use the notation $\|x\|$ when the dimension is clear and unambiguous.
- $\mathrm{tr}(M)$ denotes the trace of a matrix $M$.
- $\|M\|_{\mathcal{F}}$ stands for the Frobenius norm of a matrix $M$, i.e. $\|M\|_{\mathcal{F}} = \sqrt{\mathrm{tr}(M^T M)}$.
- $\|M\|_*$ stands for the nuclear norm of a matrix $M$, i.e. $\|M\|_* = \mathrm{tr}((M^T M)^{\frac{1}{2}})$.
- The notation $\boldsymbol{\sigma}(M)$ denotes the vector of singular values of the matrix $M$.
- $\mathrm{Id}_d$ is the identity matrix of size $d \times d$.
- For any $x \in \mathbb{R}^d$, $\mathrm{diag}(x)$ denotes the matrix of size $d \times d$ with diagonal vector $x$.
- $\widetilde{I}_d$ stands for any matrix of size $d \times d$ of the form $\mathrm{diag}((\pm 1)_{1 \le i \le d})$
- Suppose $d \ge d'$. For any matrix $M$ of size $d \times d$, we denote by $M^{(d')}$ the submatrix of size $d' \times d'$ containing the $d'$ first rows and the $d'$ first columns of $A$.
- Let $r \le d$ and $s \le d'$. For any matrix $M$ of size $r \times s$, we denote by $M^{[d,d']}$ the matrix of size $d \times d'$ of the form $\begin{pmatrix} M & 0 \\ 0 & 0 \end{pmatrix}$. When $d = d'$, we will write $M^{[d]}$.
- We use the notation $\mathbb{S}^d$ for the set of symmetric matrices of size $d \times d$, $\mathbb{S}^d_+$ the set of symmetric positive semi-definite matrices, and $\mathbb{S}^d_{++}$ the set of symmetric positive definite matrices.
- $\mathbb{1}_{d',d} = (1)_{\substack{1 \le i \le d' \\ 1 \le j \le d}}$ denotes the matrix of ones with $d'$ rows and $d$ columns.
- The notation $X \sim \mu$ means that $X$ is a random variable with probability distribution $\mu$.
- If $\mu$ is a positive measure on $\mathcal{X}$ and $\phi \colon \mathcal{X} \to \mathcal{Y}$ is a mapping, $\phi_{\#}\mu$ stands for the push-forward measure of $\mu$ by $\phi$, i.e. the measure on $\mathcal{Y}$ such that for any measurable set $\mathsf{A}$ of $\mathcal{Y}$, $\phi_{\#}\mu(\mathsf{A}) = \mu(\phi^{-1}(\mathsf{A}))$.
- If $\mu$ is a positive measure on $\mathcal{X}$, $\mathrm{supp}(\mu)$ denotes its support, i.e. the subset of $\mathcal{X}$ defined as $\mathrm{supp}(\mu) = \{x \in \mathcal{X} \mid \text{for all open set } N_x \text{ such that } x \in N_x, \ \mu(N_x) > 0\}$.
- If $X$ and $Y$ are random vectors on $\mathbb{R}^d$ and $\mathbb{R}^{d'}$, we use the notation $\mathrm{Cov}(X, Y)$ for the matrix of size $d \times d'$ of the form $\mathbb{E}\left[(X - \mathbb{E}[X])(Y - \mathbb{E}[Y])^T\right]$.
- For any positive measure $\mu$, we denote by $\bar{\mu}$ its associated centered measure, i.e. the measure such that if $X \sim \mu$, we have $X - \mathbb{E}_{X \sim \mu}[X] \sim \bar{\mu}$.
- For any $m \in \mathbb{R}^d$ and any $\Sigma \in \mathbb{S}^d_+$, we denote by $\mathrm{N}(m, \Sigma)$ the Gaussian measure of mean $m$ and covariance matrix $\Sigma$.
- For $x \in \mathcal{X}$, $\delta_x$ denotes the Dirac distribution at $x$.

## 2 Background : Mixture-Wasserstein and Gromov-Wasserstein-type distances

We recall in this section the definitions and some important properties of the different OT distances used throughout the paper. For any Polish space $\mathcal{X}$, we write $\mathcal{P}(\mathcal{X})$ the set probability measures on $\mathcal{X}$. For $d \ge 1$ and $p \ge 1$, the Wasserstein space $\mathcal{W}_p(\mathbb{R}^d)$ is defined as the set of probability measures $\mu$ on $\mathbb{R}^d$ with finite moment of order $p$, i.e. such that

$$\int_{\mathbb{R}^m} \|x\|^p \mathrm{d}\mu(x) < +\infty \,,$$

with $\|.\|$ being the Euclidean norm on $\mathbb{R}^d$.

### 2.1 Mixture-Wasserstein distance between GMMs

We present here the distance introduced in Delon and Desolneux (2020), as well as some results that will be useful in the rest of the paper. We denote $GMM_K(\mathbb{R}^d)$ the set of Gaussian mixtures on $\mathbb{R}^d$ with less than $K$ components, i.e. the set of measures in $\mathcal{P}(\mathbb{R}^d)$ which can be written

$$\mu = \sum_{k=1}^{K'} a_k \mu_k \,,$$

where $K' \le K$, $a = (a_1, \ldots, a_{K'})^T$ is in $\Delta_{K'}$, and $\{\mu_k\}_k$ is a family of pairwise distinct Gaussian distributions, each of mean $m_k \in \mathbb{R}^d$ and covariance matrix $\Sigma_k \in \mathbb{S}^d_+$. Again, to avoid degeneracy issues where locations

with no mass are accounted for, we will assume that the elements of $a$ are all positive. The set of all finite Gaussian mixture distributions on $\mathbb{R}^d$ is then written

$$GMM_\infty(\mathbb{R}^d) = \bigcup_{K \geq 0} GMM_K(\mathbb{R}^d) \ .$$

Note that the condition that the Gaussian components are pairwise distinct ensures the identifiability of the elements of $GMM_\infty(\mathbb{R}^d)$ (Yakowitz and Spragins, 1968), in the sense that two GMMs $\mu = \sum_k^K a_k \mu_k$ and $\nu = \sum_l^L b_l \nu_l$ are equal if and only if $K = L$, and we can reorder the indices such that for all $k$, $a_k = b_k$ and $\mu_k = \nu_k$. It can been shown that $GMM_\infty(\mathbb{R}^d)$ is dense in $\mathcal{W}_p(\mathbb{R}^d)$ for the metric $W_p$, meaning that any measure in $\mathcal{W}_p(\mathbb{R}^d)$ can be approximated with any precision for the distance $W_p$ by a finite Gaussian mixture distribution. Let $\mu \in GMM_K(\mathbb{R}^d)$ and $\nu \in GMM_L(\mathbb{R}^d)$. The Mixture-Wasserstein distance of order 2 is defined as

$$MW_2(\mu, \nu) = \left( \inf_{\pi \in \Pi(\mu,\nu) \cap GMM_\infty(\mathbb{R}^{2d})} \int_{\mathbb{R}^d \times \mathbb{R}^d} \|x - y\|^2 \mathrm{d}\pi(x,y) \right)^{\frac{1}{2}} \ . \tag{$MW_2$}$$

As for $W_2$ with $\mathcal{W}_2(\mathbb{R}^d)$, $MW_2$ defines a metric on $GMM_\infty(\mathbb{R}^d)$ (Delon and Desolneux, 2020). In general, the transportation plan solution of the $W_2$ problem is not a Gaussian mixture, thus by restricting the set of admissible couplings, we most of the time have $MW_2(\mu, \nu) > W_2(\mu, \nu)$. It can be shown that the difference between $MW_2(\mu, \nu)$ and $W_2(\mu, \nu)$ is upper-bounded by a term that only depends on the weights and the covariances matrices of the components of the two mixtures. An important property of $MW_2$ is that it can be written in an equivalent form, which had already been introduced in Chen et al. (2018): if $\mu = \sum_k^K a_k \mu_k$ and $\nu = \sum_l^L b_l \nu_l$, then

$$MW_2^2(\mu, \nu) = \inf_{\omega \in \Pi(a,b)} \sum_{k,l} \omega_{k,l} W_2^2(\mu_k, \nu_l) \ , \tag{2}$$

where $a = (a_1, \ldots, a_K)^T$, $b = (b_1, \ldots, b_L)^T$. From a computational point of view, this latter formulation reduces the problem to a simple small-scale discrete optimal transport problem since the $W_2$ distance between Gaussian distributions has a closed form: indeed, recall that if $\mu_k = \mathrm{N}(m_k, \Sigma_k)$ and $\nu_l = \mathrm{N}(m_l, \Sigma_l)$, then

$$W_2^2(\mu_k, \nu_l) = \|m_k - m_l\|^2 + \mathrm{tr}\left( \Sigma_k + \Sigma_l - 2\left( \Sigma_l^{\frac{1}{2}} \Sigma_k \Sigma_l^{\frac{1}{2}} \right)^{\frac{1}{2}} \right) \ . \tag{3}$$

Finally, the respective solutions $\pi^*$ and $\omega^*$ of Problems ($MW_2$) and (2) are linked by the following relationship, for all $(x, y) \in \mathbb{R}^d \times \mathbb{R}^d$,

$$\pi^*(x,y) = \sum_{k,l} \omega_{k,l}^* p_{\mu_k}(x) \delta_{y=T_{W_2}^{k,l}(x)} \ , \tag{4}$$

where $T_{W_2}^{k,l}$ is the optimal $W_2$ transport map between $\mu_k$ and $\nu_l$ and $p_{\mu_k}$ is the probability density function of $\mu_k$.

## 2.2 Gromov-Wasserstein distance

The Gromov-Wasserstein problem (Mémoli, 2011) can be defined as the following: given two *network measure spaces*, i.e. triplets of the form $(\mathcal{X}, c_\mathcal{X}, \mu)$ where $\mathcal{X}$ is a Polish space, $c_\mathcal{X} : \mathcal{X} \times \mathcal{X} \to \mathbb{R}$ is a measurable function and $\mu \in \mathcal{P}(\mathcal{X})$, it aims at finding

$$GW_p((\mathcal{X}, c_\mathcal{X}, \mu), (\mathcal{Y}, c_\mathcal{Y}, \nu)) = \left( \inf_{\pi \in \Pi(\mu,\nu)} \int_{\mathcal{X} \times \mathcal{Y}} \int_{\mathcal{X} \times \mathcal{Y}} |c_\mathcal{X}(x,x') - c_\mathcal{Y}(y,y')|^p \mathrm{d}\pi(x,y) \mathrm{d}\pi(x',y') \right)^{\frac{1}{p}} \ , \tag{5}$$

with $p \geq 1$. The fundamental metric properties of $GW_p$ have been studied in depth in (Mémoli, 2011; Sturm, 2012; Chowdhury and Mémoli, 2019). When $c_\mathcal{X}$ and $c_\mathcal{Y}$ are powers of the metrics $d_\mathcal{X}$ and $d_\mathcal{Y}$ of the base spaces $\mathcal{X}$ and $\mathcal{Y}$, $GW_p$ induces a metric over the space of *metric measure spaces* (i.e. the triplets $(\mathcal{X}, d_\mathcal{X}, \mu)$) quotiented by the *strong isomorphisms* (Sturm, 2012), where one says that two metric measure spaces $(\mathcal{X}, d_\mathcal{X}, \mu)$ and $(\mathcal{Y}, d_\mathcal{Y}, \mu)$ are strongly isomorphic if there exists an isometric bijection $\phi \colon \mathrm{supp}(\mu) \to \mathrm{supp}(\nu)$ that transports $\mu$ into $\nu$. When $c_\mathcal{X}$ and $c_\mathcal{Y}$ are not powers of the metrics of the base spaces, $GW_p$ still defines

a metric, but this time over the space of network measure spaces quotiented by the *weak isomorphisms* (Chowdhury and Mémoli, 2019), which are spaces isomorphic for the costs $c_{\mathcal{X}}$ and $c_{\mathcal{Y}}$ relatively to a third space, see (Chowdhury and Mémoli, 2019) for details. Note that in both cases, the metric property of $GW_p$ stricly holds only when it takes finite values, and so it is natural to restrict it to the following space

$$\mathbb{M}_p = \{(\mathcal{X}, c_{\mathcal{X}}, \mu) \ : \ \int_{\mathcal{X} \times \mathcal{X}} c_{\mathcal{X}}^p(x, x') \mathrm{d}\mu(x) \mathrm{d}\mu(x') < +\infty \} \ .$$

Note that when $\mathcal{X}$ and $\mathcal{Y}$ are fixed as well as $c_{\mathcal{X}}$ and $c_{\mathcal{Y}}$, it is natural to see $GW_p$ as a distance between the two measures $\mu$ and $\nu$ rather than a distance between the two network measure spaces $(\mathcal{X}, c_{\mathcal{X}}, \mu)$ and $(\mathcal{Y}, c_{\mathcal{Y}}, \nu)$. Therefore, we will denote in that case - with a slight abuse of notations - $GW_p(\mu, \nu)$ instead of $GW_p((\mathcal{X}, c_{\mathcal{X}}, \mu), (\mathcal{Y}, c_{\mathcal{Y}}, \nu))$. Finally, in the discrete setting, given $a = (a_1, \ldots, a_m)^T$ and $b = (b_1, \ldots, b_n)^T$ being respectively in $\Delta_m$ and $\Delta_n$, and given two non-negative cost matrices $C^x$ and $C^y$ of respective size $m \times m$ and $n \times n$, the Gromov-Wasserstein distance can be written as

$$GW_p(a, b) = \inf_{\omega \in \Pi(a,b)} \sum_{i,j,k,l} |C_{i,k}^x - C_{j,l}^y|^p \omega_{i,j} \omega_{k,l} \ .$$

### 2.3 Other invariant distances

Sturm (2006) has introduced another distance between metric measures spaces which takes the following form

$$D_p((\mathcal{X}, d_{\mathcal{X}}, \mu), (\mathcal{Y}, d_{\mathcal{Y}}, \nu)) = \inf_{\mathcal{Z}, \psi, \phi} W_p(\psi_\# \mu, \phi_\# \nu) \ , \tag{6}$$

where $(\mathcal{X}, d_{\mathcal{X}}, \mu)$ and $(\mathcal{Y}, d_{\mathcal{Y}}, \nu)$ are two metric measure spaces as defined in Section 2.2, $\mathcal{Z}$ is a third Polish space, and where $\psi \colon \mathcal{X} \to \mathcal{Z}$ and $\phi \colon \mathcal{Y} \to \mathcal{Z}$ are two isometric mappings. More recently, Alvarez-Melis et al. (2019) have introduced another family of invariant OT distances in the Euclidean setting which can also be used to compare distributions on spaces of different dimensions. Initially, Alvarez-Melis et al. (2019) have introduced this OT distance in the setting where $\mu$ and $\nu$ are both living in the same Euclidean space $\mathbb{R}^d$. Yet, it generalizes well to settings where $\mu$ and $\nu$ are living in spaces of different dimensions. Between two measures $\mu$ and $\nu$ on $\mathbb{R}^d$ and $\mathbb{R}^{d'}$, this reads as

$$IW_{2,\mathcal{H}}(\mu, \nu) = \left( \inf_{\pi \in \Pi(\mu,\nu)} \inf_{h \in \mathcal{H}} \int_{\mathbb{R}^d \times \mathbb{R}^{d'}} \|x - h(y)\|^2 \mathrm{d}\pi(x, y) \right)^{\frac{1}{2}} \ , \tag{$IW_2$}$$

where $\mathcal{H}$ is a class of mappings from $\mathbb{R}^{d'}$ to $\mathbb{R}^d$ encoding the invariance. This is a *non-convex* optimization problem in $\pi$ and $h$ that becomes convex in $\pi$ if $h$ is fixed and becomes also convex in $h$ if $\pi$ is fixed and $\mathcal{H}$ is a convex set. When $d$ is equal to $d'$ and both measures are centered, Alvarez-Melis et al. (2019) have notably shown that when $\nu$ is such that $\mathbb{E}_{Y \sim \nu}[YY^T] = \mathrm{Id}_d$ and when $\mathcal{H} = \mathcal{H}_1 := \{P \in \mathbb{R}^{d \times d} \ : \ \|P\|_{\mathcal{F}} \leq \sqrt{d}\}$, Problem ($IW_2$) is equivalent to the Gromov-Wasserstein problem (5) of order 2 with inner-product costs. Indeed, it can be shown that both problems are equivalent in that case to

$$\sup_{\pi \in \Pi(\mu,\nu)} \left\| \int_{\mathbb{R}^d \times \mathbb{R}^d} xy^T \mathrm{d}\pi(x, y) \right\|_{\mathcal{F}} \ , \tag{$\mathcal{F}$-COV}$$

where for any matrix $A$ of size $d \times d$, $\|A\|_{\mathcal{F}}$ denotes the Frobenius norm, i.e. $\sqrt{\mathrm{tr}(A^T A)}$. Another interesting case is when $\mathcal{H} = \mathcal{H}_2 := \mathbb{O}(\mathbb{R}^d) := \{P \in \mathbb{R}^{d \times d} \ : \ P^T P = \mathrm{Id}_d\}$ is the set of orthogonal matrices of size $d \times d$. In that case, Problem ($IW_2$) is equivalent to

$$\sup_{\pi \in \Pi(\mu,\nu)} \left\| \int_{\mathbb{R}^d \times \mathbb{R}^d} xy^T \mathrm{d}\pi(x, y) \right\|_* \ , \tag{$*$-COV}$$

where for any matrix $A$ of size $d \times d$, $\|A\|_*$ is the nuclear norm of $A$, i.e. $\|A\|_* = \mathrm{tr}((A^T A)^{\frac{1}{2}})$. Note that both Problems ($\mathcal{F}$-COV) and ($*$-COV) are *non-convex*. These results have been shown by Alvarez-Melis et al. (2019) in the case where $\mu$ and $\nu$ are discrete but can easily be extended to continuous distributions. Observe that problem ($*$-COV) consists in maximizing the sum of the singular values of the cross-covariance matrix $\int xy^T \mathrm{d}\pi(x, y)$, whereas the Problem ($\mathcal{F}$-COV) consists in maximizing the sum of the squared singular values of the cross-covariance matrix. In general, these two problems are not equivalent despite being structurally similar, as the example of Figure 1 illustrates it.

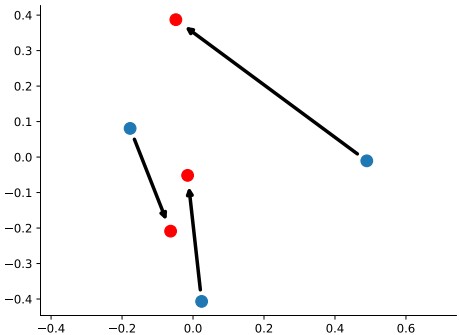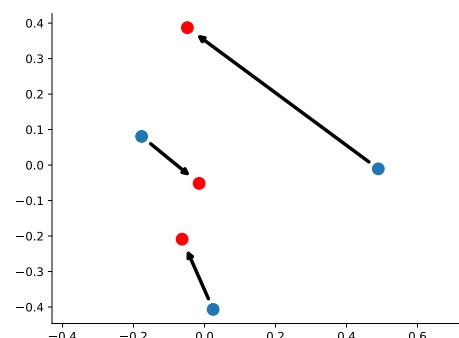

Figure 1: Transport plans between two discrete centered distributions on $\mathbb{R}^2$ composed of three points. Left: optimal coupling given by the maximization of Problem ($\mathcal{F}$-COV). Right: optimal coupling given by the maximization of Problem ($*$-COV).

## 3 Gromov-Wasserstein distance between mixture of Gaussians

In this section, we define a Gromov-Wasserstein type distance between Gaussian mixture distributions. This distance is a natural "Gromovization" of Problem (2). Indeed, as it has already been observed in the literature (Chen et al., 2018; Lambert et al., 2022), any Gaussian mixture in dimension $d$ can be identified with a probability distribution on $\mathbb{R}^d \times \mathbb{S}_+^d$, i.e. the product space of means and covariance matrices. Equivalently, a finite Gaussian mixture can be seen as a discrete probability distribution on the space of Gaussian distributions $\mathcal{N}(\mathbb{R}^d)^3$, which has been proven to be a complete metric space when endowed with $W_2$ (Takatsu, 2010) and is furthermore separable since it is a subspace of $\mathcal{W}_2(\mathbb{R}^d)$ which is itself a separable metric space when endowed with $W_2$ (Bolley, 2008). Since the theory of optimal transport still applies on measures over non-Euclidean spaces (Villani, 2008), it follows that Problem (2) can formally be thought as a simple OT problem between two discrete measures in $\mathcal{P}(\mathcal{N}(\mathbb{R}^d))$. Thus, one can define directly its Gromov version.

**Definition 1.** *Let* $\mu = \sum_k a_k \mu_k$ *and* $\nu = \sum_l b_l \nu_l$ *be two Gaussian mixtures respectively on* $\mathbb{R}^d$ *and* $\mathbb{R}^{d'}$, *we define*

$$MGW_2^2(\mu, \nu) = \inf_{\omega \in \Pi(a,b)} \sum_{i,j,k,l} |W_2^2(\mu_i, \mu_k) - W_2^2(\nu_j, \nu_l)|^2 \omega_{i,j} \omega_{k,l} . \qquad (MGW_2)$$

Unlike $MW_2$, there is no straightforward equivalent continuous formulation of this latter problem. In particular, it is not clear whether Problem ($MGW_2$) is equivalent or not to the continuous GW problem between $\mu$ and $\nu$ - seen as continuous measures on $\mathbb{R}^d$ and $\mathbb{R}^{d'}$ - where the set of admissible couplings is restricted to Gaussian mixture distributions. Thanks to the identifiability property of the set of finite Gaussian mixtures, we have that each $\mu \in GMM_\infty(\mathbb{R}^d)$ is associated with a unique discrete distribution $\tilde{\mu} \in \mathcal{P}(\mathcal{N}(\mathbb{R}^d))$ and $MGW_2$ between $\mu$ and $\nu$ coincides with $GW_2$ with squared $W_2$ costs between their associated discrete measures $\tilde{\mu}$ and $\tilde{\nu}$ in $\mathcal{P}(\mathcal{N}(\mathbb{R}^d))$. Finally, note that we have defined $MGW_2$ only between GMMs with finite number of components because there is in general no identifiability property for infinite Gaussian mixtures. As an outcome, for a given infinite GMM $\mu$ on $\mathbb{R}^d$, there might be more than one associated continuous measure $\tilde{\mu}$ on $\mathcal{N}(\mathbb{R}^d)$. For instance, the standard Normal distribution $N(0,1)$ can naturally be identified in $\mathcal{P}(\mathcal{N}(\mathbb{R}))$ with the Dirac distribution at $N(0,1)$, but also with the Normal distribution $N(0,1/2)$ over the parametrized line $\{N(\theta, 1/2) \in \mathcal{N}(\mathbb{R}) : \theta \in \mathbb{R}\}$, or with $N(0,1)$ over the parametrized line $\{\delta_\theta \in \mathcal{N}(\mathbb{R}) : \theta \in \mathbb{R}\}$.

---

[3]$\mathcal{N}(\mathbb{R}^d)$ includes the degenerate Gaussian distributions, as for instance the Dirac distributions.

### 3.1 Metric properties

Here we study the metric property of $MGW_2$ that mainly arises from the Gromov-Wasserstein structure of Problem ($MGW_2$). Indeed, the following result is a direct consequence of the theory developed by Sturm (2012).

**Proposition 2.** *In the following, $\mu = \sum_k a_k \mu_k$ and $\nu = \sum_l b_l \nu_l$ are two GMMs respectively in $GMM_K(\mathbb{R}^d)$ and $GMM_L(\mathbb{R}^{d'})$.*

*(i) $MGW_2$ is non-negative and symmetric.*

*(ii) $MGW_2$ satisfies the triangle inequality, i.e. for any $\xi \in GMM_S(\mathbb{R}^{d''})$,*

$$MGW_2(\mu, \nu) \leq MGW_2(\mu, \xi) + MGW_2(\xi, \nu) .$$

*(iii) $MGW_2(\mu, \nu) = 0$ if and only if there exists a bijection $\phi \colon \{\mu_k\}_k \to \{\nu_l\}_l$ such that $\nu = \sum_k a_k \phi(\mu_k)$ and $\phi$ is an isometry for $W_2$, i.e. for all $k$ and $i$ smaller than $K$, $W_2(\phi(\mu_k), \phi(\mu_i)) = W_2(\mu_k, \mu_i)$.*

*Sketch of proof.* The proof of these results mainly consists in applying the theory of Sturm (2012), using the facts that $\mathcal{N}(\mathbb{R}^d)$ is complete (Takatsu, 2010), separable (Bolley, 2008), and metrizable with the Wasserstein distance. See Appendix B.1 for the full proof. □

$MGW_2$ defines thus a pseudometric on the set of all finite Gaussian mixtures of arbitrary dimensions, i.e. the set,

$$\mathcal{GMM}_\infty = \bigsqcup_{d \geq 1} GMM_\infty(\mathbb{R}^d) ,$$

that is invariant to the mappings $\phi$ that transform a finite Gaussian mixture $\sum_{k=1} a_k \mu_k$ into another finite Gaussian mixture of the form $\sum_{k=1}^K a_k \nu_k$ such that for all $k$ and $i$ smaller than $K$, $W_2(\nu_k, \nu_i) = W_2(\mu_k, \mu_i)$. A question that arises is: are all these mappings $\phi$ between $GMM_\infty(\mathbb{R}^{d'})$ and $\mathcal{P}(\mathbb{R}^d)$ always associated with mappings $T \colon \mathbb{R}^{d'} \to \mathbb{R}^d$ that are isometries for the Euclidean norm and such that $T_{\#}\mu$ coincides with $\phi(\mu)$ for every $\mu \in GMM_\infty(\mathbb{R}^{d'})$? We can already state the following converse result.

**Proposition 3.** *Let $d \geq d'$, and let $T \colon \mathbb{R}^{d'} \to \mathbb{R}^d$ be a mapping that is an isometry for the Euclidean norm. Then the mapping $\phi_T \colon GMM_\infty(\mathbb{R}^{d'}) \to \mathcal{P}(\mathbb{R}^d)$ defined as $\phi_T(\mu) = T_{\#}\mu$ for all $\mu \in GMM_\infty(\mathbb{R}^{d'})$, is such that for any $\mu$ of the form $\Sigma_{k=1}^K a_k \mu_k$, $\phi_T(\mu)$ is in $GMM_\infty(\mathbb{R}^{d'})$ and is of the form $\Sigma_{k=1}^K a_k \nu_k$, with $\{\nu_k\}_{k=1}^K$ being such that, for all $k$ and $i$ smaller than $K$, $W_2(\nu_k, \nu_i) = W_2(\mu_k, \mu_i)$, and so $MGW_2(\mu, T_{\#}\mu) = 0$.*

*Sketch of proof.* The proof of this result mainly consists in showing that for any $\mu \in GMM_\infty(\mathbb{R}^{d'})$, $\phi_T(\mu)$ is in $GMM_\infty(\mathbb{R}^d)$ because $T$ is necessarily affine, as a direct consequence of the Mazur-Ulam theorem (Mazur and Ulam, 1932) which implies that any isometry from $\mathbb{R}^{d'}$ to $\mathbb{R}^d$ (endowed with the Euclidean norm) is necessarily affine. See Appendix B.2 for the full proof. □

Hence, if $T \colon \mathbb{R}^{d'} \to \mathbb{R}^d$ is an isometry for the Euclidean norm, then $MGW_2$ is invariant to the mapping $\phi_T \colon GMM_\infty(\mathbb{R}^{d'}) \to GMM_\infty(\mathbb{R}^d)$ given for all $\mu \in GMM_\infty(\mathbb{R}^{d'})$, by $\phi_T(\mu) = T_{\#}\mu$. Yet, in general, there exist mappings $\phi \colon \mathcal{W}_2(\mathbb{R}^{d'}) \to \mathcal{W}_2(\mathbb{R}^d)$ that are isometries for $W_2$ and that are not induced by any mapping $T \colon \mathbb{R}^{d'} \to \mathbb{R}^d$ that is an isometry for the Euclidean norm. This has been proven by Kloeckner (2010) in the general case when considering isometries defined all over $\mathcal{W}_2(\mathbb{R}^d)$, but remains true when restricting to isometries defined over subspaces of $\mathcal{N}(\mathbb{R}^d)$ as the following example suggests.

**Example 4.** *let $\mathcal{N}_{++}(\mathbb{R})$ be the set of one-dimensional Gaussian distributions with strictly positive mean. Let $\phi \colon \mathcal{N}_{++}(\mathbb{R}) \to \mathcal{N}_{++}(\mathbb{R})$ be the mapping that swaps the mean and the standard deviation, i.e. such that for any $\gamma = \mathrm{N}(m_\gamma, \sigma_\gamma^2)$ with $m_\gamma > 0$ and $\sigma_\gamma > 0$, $\phi(\gamma) = \mathrm{N}(\sigma_\gamma, m_\gamma^2)$. Then $\phi$ is an isometry for $W_2$. Observe indeed that for $\gamma$ and $\zeta$ in $\mathcal{N}_{++}(\mathbb{R})$, we have*

$$W_2(\phi(\gamma), \phi(\zeta)) = (\sigma_\gamma - \sigma_\zeta)^2 + (m_\gamma - m_\zeta)^2 = W_2(\gamma, \zeta) .$$

Thus $\phi$ is an isometry for $W_2$, yet $\phi$ is not induced by any isometry of $\mathbb{R}$. Hence there exist mappings from $GMM_\infty(\mathbb{R}^{d'})$ to $GMM_\infty(\mathbb{R}^d)$ that satisfy the conditions above but which are not induced by isometries for the Euclidean norm from $\mathbb{R}^{d'}$ to $\mathbb{R}^d$.

### 3.2 $MGW_2$ in practice

**Using $MGW_2$ on discrete data distributions.** Most applications of optimal transport involve discrete data that can be thought as samples drawn from underlying distributions, which are not GMMs in general. In those applications, we aim to evaluate an OT distance between two distributions of the form $\hat{\mu} = (1/M) \sum_i \delta_{x_i}$ and $\hat{\nu} = (1/N) \sum_j \delta_{y_j}$ where $\{x_i\}_i$ and $\{y_j\}_j$ are families of respectively $M$ and $N$ vectors of $\mathbb{R}^d$ and $\mathbb{R}^{d'}$. Though $\hat{\mu}$ and $\hat{\nu}$ can be thought as mixtures of degenerate Gaussian distributions, evaluating directly $MGW_2(\hat{\mu}, \hat{\nu})$ is not particularly interesting since we have in that case $MGW_2(\hat{\mu}, \hat{\nu}) = GW_2(\|.\|^2, \|.\|^2, \hat{\mu}, \hat{\nu})$. However, we can design a pseudometric $MGW_{K,2}$ between $\hat{\mu}$ and $\hat{\nu}$ by fitting two GMMs $\mu$ and $\nu$ with $K$ components on $\hat{\mu}$ and $\hat{\nu}$ and then setting $MGW_{K,2}(\hat{\mu}, \hat{\nu}) = MGW_2(\mu, \nu)$. The approximation of $\hat{\mu}$ and $\hat{\nu}$ by $\mu$ and $\nu$ can be done by maximizing the log-likelihood of the GMMs with the EM algorithm (Dempster et al., 1977). Note that if $K$ is chosen too small, the approximations of $\hat{\mu}$ and $\hat{\nu}$ will be of bad quality and we are likely to observe undesirable behaviors, as for instance having $MGW_{K,2}(\hat{\mu}, \hat{\nu}) = 0$ despite $\hat{\mu}$ and $\hat{\nu}$ not being equal up to an isometry. Thus, the choice of $K$ must be a compromise between the quality of the approximation given by the GMM and the computational cost. To illustrate the pratical use of $MGW_2$ on a simple toy example, we draw 150 samples from the spiral dataset provided in the scikit-learn toolbox[4] (Pedregosa et al., 2011) and we apply rotations with various angles on this dataset. We then fit independently GMMs with 20 components on the initial and the target rotated datasets and we compute $MGW_2$ between the two obtained GMMs. We also compute $GW_2$ with inner-product as cost functions, $MW_2$ using also 20 Gaussian components and $W_2$. The results can be found in Figure 2. As expected, $MGW_2$ is rotation-invariant as $GW_2$ which is not the case of $MW_2$ and $W_2$.

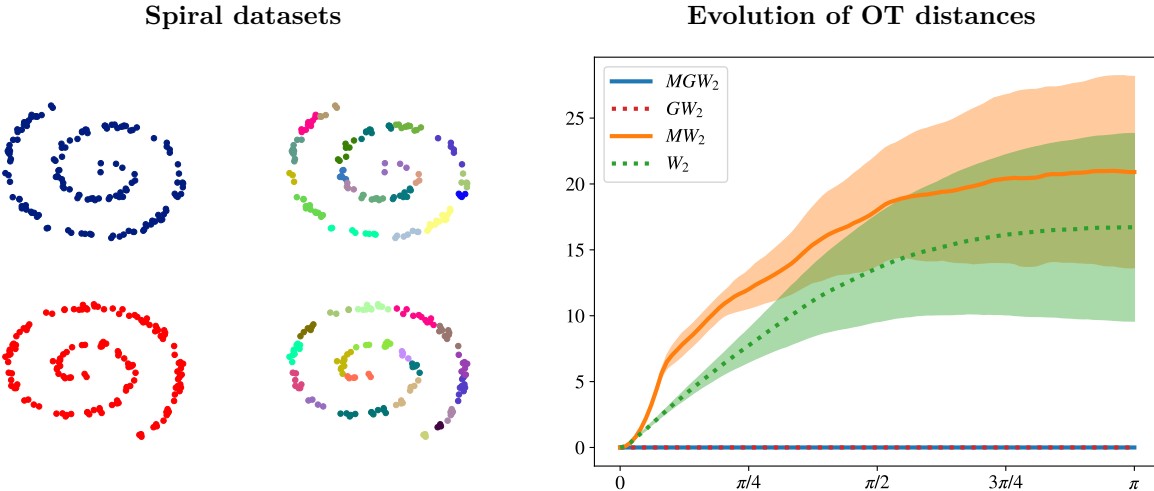

Figure 2: Left first column: spiral datasets (in blue and red) composed of 150 points of $\mathbb{R}^2$. The red dataset corresponds to points sampled from the distribution of the blue dataset rotated from $\pi$. Left second column: The two corresponding learned GMMs with 20 components via EM algorithm (each color corresponds to a Gaussian component of the GMMs). Right: evolution of $MGW_2$, $GW_2$, $MW_2$, and $W_2$ between the initial distribution (in blue) and the rotated ones in function of the angle of rotation. Experiments are averaged over 10 runs and the colored bands correspond to $+/-$ the standard deviation. This experiment is inspired from Vayer et al. (2019b).

---

[4]The package is accessible here: https://scikit-learn.org/stable/.

**Difficulty of designing a transportation plan.** The $MGW_2$ problem can be used on discrete data to provide an optimal coupling $\omega^*$ between the Gaussian components of the two Gaussian mixtures $\mu$ and $\nu$ that approximate the discrete data distributions $\hat{\mu}$ and $\hat{\nu}$. However, some applications require an coupling $\pi$ between the points that compose $\hat{\mu}$ and $\hat{\nu}$. It is not straightforward to derive such a transportation plan $\pi$ from the plan $\omega^*$ that minimizes the $MGW_2$ problem. A naive heuristic approach would be to define $\pi$ in a similar way to (4), transporting the Gaussian components using restricted-$GW_2$ transport maps (Delon et al., 2022a) instead of $W_2$ transport maps. Yet this approach introduces too many degrees of freedom as it consists in transporting the Gaussian components independently, without taking into account the global structure of the mixture, see Figure 3 for an illustrative example. Since selecting the solution that preserves the global structure of the mixture among all the candidates seems to be a difficult combinatorial problem, a better solution to design such plan would be to derive explicitly the isometric transformation that has been implicitly applied to one of the two measures when solving the $MGW_2$ problem. This is the idea behind the embedded Wasserstein distance that we introduce in the following section.

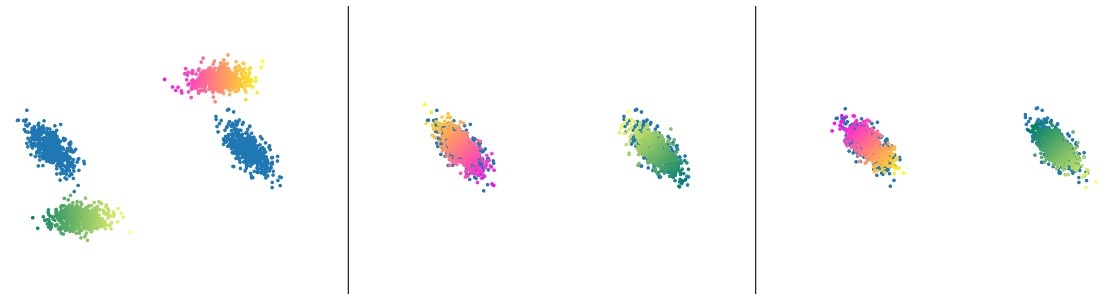

Figure 3: Left: two discrete distributions $\hat{\mu}$ (in gradient of colors) and $\hat{\nu}$ (in blue) that have been drawn from two GMMs. The colors have been added to $\hat{\mu}$ in order to visualize the couplings between $\hat{\mu}$ and $\hat{\nu}$. Middle and right: two possible solutions of transport of $\hat{\mu}$ obtained by plugging the discrete plan that minimizes $MGW_2$ in (4), using restricted-$GW_2$ transport maps (Delon et al., 2022a) to transport the Gaussian components. Observe that the middle solution preserves the global structure of the mixture, in the sense that points that are close to each other but associated with different Gaussian components remain close when tranported. This is not the case for the right solution.

## 4 Embedded Wasserstein distance

In this section, we define an alternative distance to Gromov-Wasserstein also invariant to isometries which specifies the isometric transformation applied to one of the measure when computing the distance.

**Definition 5.** *Let $\mu \in \mathcal{P}(\mathbb{R}^d)$ and $\nu \in \mathcal{P}(\mathbb{R}^{d'})$. For $r \geq 1$ and $s \geq 1$, let us denote $\mathrm{Isom}_s(\mathbb{R}^r)$ the set of all isometries - for the Euclidean norm - from $\mathbb{R}^s$ to $\mathbb{R}^r$. We define*

$$EW_2(\mu, \nu) = \inf \left\{ \inf_{\phi \in \mathrm{Isom}_{d'}(\mathbb{R}^d)} W_2(\mu, \phi_\# \nu), \inf_{\psi \in \mathrm{Isom}_d(\mathbb{R}^{d'})} W_2(\psi_\# \mu, \nu) \right\}, \qquad (EW_2)$$

*with the convention that the infimum over an empty set is equal to $+\infty$.*

Observe that if $d > d'$, the set $\mathrm{Isom}_d(\mathbb{R}^{d'})$ is empty and so $EW_2(\mu, \nu) = \inf_{\phi \in \mathrm{Isom}_{d'}(\mathbb{R}^d)} W_2(\mu, \phi_\# \nu)$. In contrast, if $d < d'$, $\mathrm{Isom}_{d'}(\mathbb{R}^d)$ is empty and so $EW_2(\mu, \nu) = \inf_{\psi \in \mathrm{Isom}_d(\mathbb{R}^{d'})} W_2(\psi_\# \mu, \nu)$. When $d = d'$, the two infimums are equivalent. **In all what follows, we will suppose without loss of generality that $d \geq d'$.**

### 4.1 Properties of $EW_2$

We present here three properties of $EW_2$ which motivate its use between Gaussian mixture models. First, we start by showing that $EW_2$ defines a pseudometric that is invariant to isometries.

**Proposition 6.** *$EW_2$ defines a pseudometric on $\bigsqcup_{k \geq 1} \mathcal{W}_2(\mathbb{R}^k)$ such that for any $\mu \in \mathcal{W}_2(\mathbb{R}^d)$ and $\nu \in \mathcal{W}_2(\mathbb{R}^{d'})$, $EW_2(\mu, \nu) = 0$ if and only if there exists an isometry $\phi \colon \mathbb{R}^{d'} \to \mathbb{R}^d$ such that $\nu = \phi_{\#}\mu$.*

*Sketch of proof.* Non-negativity and symmetry are straightforward. The triangular inequality can be proved observing first that the infimum in $\phi$ is always achieved, then remarking that $EW_2$ remains unchanged when one of the two measures is immersed in a third Euclidean space of greater dimension than $d$ and $d'$. This makes $EW_2$ closely related to the distance between metric measure spaces introduced by Sturm (2006) presented in Section 2.3. See Appendix B.4 for the full proof. □

Now we show that $EW_2$ is equivalent to the OT distance introduced by Alvarez-Melis et al. (2019) described in Section 2.3 for a particular choice of transformation space $\mathcal{H}$. In all what follows, we denote $\mathbb{V}_{d'}(\mathbb{R}^d)$ the *Stiefel manifold* (James, 1976), i.e. the set of rectangular othogonal matrices of size $d \times d'$ such that $P^T P = \mathrm{Id}_{d'}$. More precisely, we show the following result.

**Proposition 7.** *Let $\mu \in \mathcal{W}_2(\mathbb{R}^d)$ and $\nu \in \mathcal{W}_2(\mathbb{R}^{d'})$ and let suppose without loss of generality $d \geq d'$. Then,*

$$EW_2^2(\mu, \nu) = \inf_{\pi \in \Pi(\mu, \nu)} \inf_{P \in \mathbb{V}_{d'}(\mathbb{R}^d), \ b \in \mathbb{R}^d} \int_{\mathbb{R}^d \times \mathbb{R}^{d'}} \|x - Py - b\|^2 \mathrm{d}\pi(x, y) . \tag{7}$$

*Moreover this latter problem is equivalent to*

$$\sup_{\pi \in \Pi(\bar{\mu}, \bar{\nu})} \left\| \int_{\mathbb{R}^d \times \mathbb{R}^{d'}} xy^T \mathrm{d}\pi(x, y) \right\|_* . \tag{$*$-COV}$$

*Sketch of proof.* Equation (7) is a consequence of (Delon et al., 2022b, Lemma 3.3) and of the Mazur-Ulam theorem (Mazur and Ulam, 1932), which implies that any isometry from $\mathbb{R}^{d'}$ to $\mathbb{R}^d$ (endowed with the Euclidean norm) is necessarily affine. The equivalence with Problem ($*$-COV) is then roughly a consequence of (Alvarez-Melis et al., 2019, Lemma 4.2) which implies that Problem (7) is achieved in $P$ at $P^* = U_\pi \mathrm{Id}_{d'}^{[d,d']} V_\pi^T$ where $U_\pi \in \mathbb{O}(\mathbb{R}^d)$ and $V_\pi \in \mathbb{O}(\mathbb{R}^{d'})$ are the left and right orthogonal matrices associated with the Singular Value Decomposition (SVD) of $\int xy^T \mathrm{d}\pi(x, y)$. See Appendix B.3 for the full proof. □

Since Problem ($*$-COV) is in general not equivalent to Problem ($\mathcal{F}$-COV), the $EW_2$ problem is in general not equivalent to the $GW_2$ problem with inner-product costs. However, the following result shows that between Gaussian distributions, the two problems share some common solutions.

**Proposition 8.** *Suppose without loss of generality that $d \geq d'$. Let $\mu = \mathrm{N}(0, \Sigma_0)$ and $\nu = \mathrm{N}(0, \Sigma_1)$ be two centered Gaussian measures on $\mathbb{R}^d$ and $\mathbb{R}^{d'}$. Let $P_0, D_0$ and $P_1, D_1$ be the respective diagonalizations of $\Sigma_0$ ($= P_0 D_0 P_0^T$) and $\Sigma_1$ ($= P_1 D_1 P_1^T$) that sort the eigenvalues in non-increasing order. We suppose that $\mu$ is not degenerate, i.e. $\Sigma_0$ is non-singular. Then the problem*

$$EW_2(\mu, \nu) = \inf_{P \in \mathbb{V}_{d'}(\mathbb{R}^d)} W_2(\mu, P_{\#}\nu) ,$$

*admits solutions of the form $(\pi^*, P^*)$ with $P^*$ of the form $P^* = P_0 \widetilde{I}_{d'}^{[d,d']} P_1^T$ and $\pi^* = (\mathrm{Id}_d, T)_{\#}\mu$ with $T$ being any affine map such that for all $x \in \mathbb{R}^d$,*

$$T(x) = P_1 \left( \widetilde{I}_{d'} D_1^{\frac{1}{2}} D_0^{(d')^{-\frac{1}{2}}} \right)^{[d',d]} P_0^T x .$$

*In other terms, the solutions of Problem (5) with inner-product costs exhibited in Delon et al. (2022a) are also solutions of Problem ($EW_2$). Furthermore,*

$$EW_2^2(\mu, \nu) = \mathrm{tr}(D_0) + \mathrm{tr}(D_1) - 2\mathrm{tr}(D_0^{(d')^{\frac{1}{2}}} D_1^{\frac{1}{2}}) .$$

*Sketch of proof.* The proof of this result is inspired from the proof of Equation (3) given by Givens et al. (1984) that is based on Lagrangian analysis. The main difference with the proof of Equation (3) lies in the introduction of an additional variable $P$ with constraint $P \in \mathbb{V}_{d'}(\mathbb{R}^d)$. See Appendix B.5 for the full proof. □

Note that it is not clear if the two problems are strictly equivalent or only share some common solutions because it is not clear, to the best of our knowledge, that the solutions exhibited above are the only solutions of the $GW_2$ problem with inner-products costs, see Delon et al. (2022a) for more details. To complete this section, we emphasize that $EW_2$ is different from the distance proposed in Cai and Lim (2022), that we call here $PW_2$ for Projection Wasserstein discrepancy. Details about this difference can be found in Appendix C.

### 4.2 Embedded Wasserstein distance between GMMs

Similarly to Delon and Desolneux (2020), one can define an OT distance derived from $EW_2$ when $\mu$ and $\nu$ are GMMs by restricting the set of admissible couplings to be themselves GMMs.

**Definition 9.** *Let $\mu \in GMM_K(\mathbb{R}^d)$ and $\nu \in GMM_L(\mathbb{R}^{d'})$ and suppose that $d \geq d'$. We define*

$$MEW_2(\mu, \nu) = \inf \left\{ \inf_{\phi \in \mathrm{Isom}_{d'}(\mathbb{R}^d)} MW_2(\mu, \phi_\# \nu), \inf_{\psi \in \mathrm{Isom}_d(\mathbb{R}^{d'})} MW_2(\psi_\# \mu, \nu) \right\} . \tag{8}$$

As before, one can reformulate this latter problem by observing that the isomorphic mappings for the Euclidean norm are necessarily of the form $Px + b$ with $P \in \mathbb{V}_{d'}(\mathbb{R}^d)$ and $b \in \mathbb{R}^d$. Similarly to $EW_2$, one can show that the infimum in $\phi$ is always achieved and that $MEW_2$ satisfies all the properties of a pseudometric on $\mathcal{GMM}_\infty$ by simply replacing $W_2$ by $MW_2$ in the proof of Proposition 6. Supposing without loss of generality that $d \geq d'$ and using the equivalent discrete formulation (2) of the $MW_2$ problem, we get that for $\mu = \sum_k a_k \mu_k$ and $\nu = \sum_l b_l \nu_l$, the problem is equivalent to

$$\inf_{P \in \mathbb{V}_{d'}(\mathbb{R}^d)} \inf_{\omega \in \Pi(a,b)} \sum_{k,l} \omega_{k,l} W_2^2(\mu_k', P_\# \nu_l') , \tag{$MEW_2$}$$

where for any $k \leq K$ and $l \leq L$, $\mu_k'$ and $\nu_l'$ are the Gaussian components respectively associated to the centered GMMs $\bar{\mu}$ and $\bar{\nu}$. Note that $\mu_k'$ and $\nu_l'$ are not necessarily themselves centered.

#### 4.2.1 Numerical solver

This time, it is not possible to derive analytically the closed form of the optimal $P^*$ for Problem ($MEW_2$). However, one can still solve the problem numerically using an alternate minimization scheme. Indeed, Problem ($MEW_2$) is not convex in $P$ and $\omega$, but is convex in $\omega$ if $P$ is fixed and is furthermore a simple small-scale discrete OT problem in that case, which motivates the use of an alternating optimization scheme for solving this problem. However, Problem ($MEW_2$) is not convex in $P$ for a fixed $\omega$ because the feasible set, i.e. the Stiefel manifold $\mathbb{V}_{d'}(\mathbb{R}^d)$, is not convex. For a fixed $\omega$, the minimization in $P$ can be done by projected gradient descent (Calamai and Moré, 1987), i.e. for a given iterate $P^{\{i\}}$ and a given $\omega$, the next iterate $P^{\{i+1\}}$ is given by

$$P^{\{i+1\}} = \kappa_{\mathbb{V}_{d'}(\mathbb{R}^d)} \left( P^{\{i\}} - \eta \frac{\partial J_\omega(P^{\{i\}})}{\partial P} \right) ,$$

where $\kappa_{\mathbb{V}_{d'}(\mathbb{R}^d)}$ is the projection mapping on the Stiefel manifold, where $\eta > 0$ and where for all matrices $P$ of size $d' \times d$, $J_\omega(P) = \sum_{k,l} \omega_{k,l} W_2(\mu_k', P_\# \nu_l')$. Observe that we have, as a byproduct of Proposition 7, that for all $P$ of size $d' \times d$, the projection $\kappa_{\mathbb{V}_{d'}(\mathbb{R}^d)}$ is written

$$\kappa_{\mathbb{V}_{d'}(\mathbb{R}^d)}(P) = U_P \, \mathrm{Id}_{d'}^{[d,d']} V_P^T ,$$

where $U_P \in \mathbb{O}(\mathbb{R}^d)$ and $V_P \in \mathbb{O}(\mathbb{R}^{d'})$ are respectively the left and right orthogonal matrices associated with the SVD of $P$. Indeed, this projection can be written

$$\kappa_{\mathbb{V}_{d'}(\mathbb{R}^d)}(P) = \underset{\tilde{P} \in \mathbb{V}_{d'}(\mathbb{R}^d)}{\arg\min} \|P - \tilde{P}\|_{\mathcal{F}}^2 = \underset{\tilde{P} \in \mathbb{V}_{d'}(\mathbb{R}^d)}{\arg\min} \left[ \|P\|_{\mathcal{F}}^2 + \|\tilde{P}\|_{\mathcal{F}}^2 - 2\mathrm{tr}(\tilde{P}^T P) \right] .$$

Since for all $\tilde{P} \in \mathbb{V}_{d'}(\mathbb{R}^d)$, $\|\tilde{P}\|_{\mathcal{F}}^2 = d'$, we get that the problem is equivalent to $\sup_{\tilde{P} \in \mathbb{V}_{d'}(\mathbb{R}^d)} \mathrm{tr}(\tilde{P}^T P)$ which is maximized when $\tilde{P} = U_P \, \mathrm{Id}_{d'}^{[d,d']} V_P^T$, see the sketch of proof of Proposition 7. Finally, this yields to

---

**Algorithm 1** Mixture Embedded Wasserstein solver

---

**Require:** $\mu = \sum_k^K a_k \mu_k$, $\nu = \sum_l^L b_l \nu_l$, $P^{\{0\}} \in \mathbb{V}_{d'}(\mathbb{R}^d)$, $\eta > 0$.
 1: **while** not converged **do**
 2:     $[C]_{k,l} \leftarrow W_2^2(\mu_k, \nu_l)$ for $k = 1, \dots, K$; $l = 1, \dots, L$
 3:     $\omega^{\{i\}} \leftarrow \text{SOLVE-OT}(a, b, C)$                    $\triangleright$ Solve a classic OT problem.
 4:     **while** not converged **do**                    $\triangleright$ Do projected gradient descent on $P$.
 5:         $A \leftarrow P^{\{i-1\}} - \eta \partial J_{\omega^{\{i\}}}(P^{\{i-1\}})/\partial P$
 6:         $U, \Sigma, V^T \leftarrow \text{SVD}(A)$
 7:         $P^{\{i\}} \leftarrow U \,\text{Id}_{d'}^{[d,d']} V^T$
 8:     **end while**
 9: **end while**
10: **return** $\omega$, $P$

---

Algorithm 1. Note that more involved optimization procedures using the specific structure of the Stiefel manifold could probably be used here (Boumal, 2023).

When $\mu$ and $\nu$ are only composed of non-degenerate Gaussian components, one can compute $\partial J_\omega(P)/\partial P$ either by using automatic differentiation (Baydin et al., 2018) or by using the following technical result, whose proof is postponed to Appendix B.

**Lemma 10.** *Let for any $1 \leq k \leq K$, $\mu_k = \mathrm{N}(m_{0k}, \Sigma_{0k})$ with $m_{0k} \in \mathbb{R}^d$ and $\Sigma_{0k} \in \mathbb{S}_{++}^d$ and for any $1 \leq l \leq L$, $\nu_l = \mathrm{N}(m_{1l}, \Sigma_{1l})$ with $m_{1l} \in \mathbb{R}^{d'}$ and $\Sigma_{1l} \in \mathbb{S}_{++}^{d'}$. For any $\omega$ in the $K \times L$ simplex, let $J_\omega : \mathbb{R}^{d \times d'} \to \mathbb{R}$ be the functional defined, for all matrix $P$ of size $d \times d'$, by*

$$J_\omega(P) = \sum_{k,l} \omega_{k,l} W_2^2(\mu_k, P_\# \nu_l) \,.$$

*Then for any full-rank matrix $P$ of size $d \times d'$, we have*

$$\frac{\partial J_\omega(P)}{\partial P} = 2 \sum_{k,l} \omega_{k,l} \left[ P m_{1l} m_{1l}^T - m_{0k} m_{1l}^T - \Sigma_{0k} P \Sigma_{1l}^{\frac{1}{2}} (\Sigma_{1l}^{\frac{1}{2}} P^T \Sigma_{0k} P \Sigma_{1l}^{\frac{1}{2}})^{-\frac{1}{2}} \Sigma_{1l}^{\frac{1}{2}} \right] \,.$$

**Initialization procedure.** Since the problem is non-convex, the solution to which Algorithm 1 converges strongly depends on the initialization of $P$. It is therefore crucial to design a good initialization procedure. To do so, we propose to use the *annealing* scheme introduced by Alvarez-Melis et al. (2019). More precisely, we propose to set the initial $P$ as the solution of the following iterative procedure. First we solve an entropic-regularized $W_2$ problem between the two discrete measures $\mu^\circ = \sum_k a_k \delta_{m_{0k}}$ and $\nu^\circ = \sum_k b_l \delta_{m_{1l}}$ with a large value of regularization $\varepsilon_0$ in order to obtain a coupling $\omega^{\{1\}}$. Then we set

$$P^{\{1\}} = \kappa_{\mathbb{V}_{d'}(\mathbb{R}^d)} \left( \sum_{k,l} \omega_{k,l}^{\{1\}} m_{0k} m_{1l}^T \right) \,.$$

We then solve another entropic-regularized $W_2$ problem, this time between $\mu^\circ$ and $P_\#^{\{1\}} \nu^\circ$, using a smaller value of regularization $\varepsilon_1 = \alpha \times \varepsilon_0$ with $\alpha \in (0,1)$. We obtain thus a new coupling $\omega^{\{2\}}$ and we can then derive $P^{\{2\}}$ as previously. We repeat this procedure $N_{it}$ times until the regularization term $\varepsilon_{N_{it}}$ becomes small enough. This boils down to Algorithm 2.

In practice, we set in all our experiments $\alpha = 0.95$ and $\varepsilon_0 = 1$ as in Alvarez-Melis et al. (2019). Furthermore we observed that in most cases, setting $N_{it} = 10$ was sufficient to obtain a good initialization of $P$ for Algorithm 1.

### 4.2.2 Transportation plans and transportation maps

Since ($MEW_2$) has a continous equivalent formulation (8), one can derive from any optimal solution $(\omega^*, P^*)$ of the former, an optimal solution $(\pi^*, \phi^*)$ of the latter. More precisely, we have on the one hand for all

---

**Algorithm 2** Annealed initialization procedure for mixture embedded Wasserstein

---

**Require:** $a$, $b$, $\{m_{0k}\}_k^K$, $\{m_{1l}\}_l^L$, $\varepsilon_0 > 0$, $\alpha \in (0,1)$, $P^{\{0\}} = \mathrm{Id}_{d'}^{[d,d']}$
  1: **for** $i = 1, \ldots, N_{it}$ **do**
  2:     $[C]_{k,l} \leftarrow \|m_{0k} - P^{\{i-1\}} m_{1l}\|^2$
  3:     $\omega^{\{i\}} \leftarrow \varepsilon\text{-OT}(a, b, C, \varepsilon_{i-1})$                     ▷ Solve a regularized OT problem.
  4:     $A \leftarrow \sum_{k,l} \omega_{k,l}^{\{i\}} m_{0k} m_{1l}^T$
  5:     $U, \Sigma, V^T \leftarrow \mathrm{SVD}(A)$
  6:     $P^{\{i\}} = U \, \mathrm{Id}_{d'}^{[d,d']} \, V^T$
  7:     $\varepsilon_i \leftarrow \alpha \varepsilon_{i-1}$                     ▷ Annealing scheme.
  8: **end for**
  9: **return** $P$

---

$y \in \mathbb{R}^{d'}$, $\phi^*(y) = P^* y + b^*$, where $b^* = \mathbb{E}_{X \sim \mu}[X] - P^* \mathbb{E}_{Y \sim \nu}[Y]$, and on the other hand for all $(x,y) \in \mathbb{R}^d \times \mathbb{R}^{d'}$,

$$\pi^*(x,y) = \sum_{k,l} \omega_{k,l}^* p_{\mu_k}(x) \delta_{y = \psi^* \circ T_{W_2}^{k,l}(x)} \, , \tag{9}$$

where $T_{W_2}^{k,l}$ is the optimal $W_2$ transport map between $\mu'_k$ and $P_\#^* \nu'_l$ (where we recall that $\mu'_k$ and $\nu'_l$ are the Gaussian components of the centered GMMs) and $\psi^* \colon \mathbb{R}^d \to \mathbb{R}^{d'}$ is defined for all $x \in \mathbb{R}^d$ as $\psi^*(x) = P^{*T}(x - b^*)$. As in Delon and Desolneux (2020), it is possible to define a unique assignment of each $x$ by setting for all $x \in \mathbb{R}^d$,

$$T_{\text{mean}}(x) = \mathbb{E}_{(X,Y) \sim \pi^*}[Y | X = x] = \frac{\sum_{k,l} \omega_{k,l}^* p_{\mu_k}(x) \psi^* \circ T_{W_2}^{k,l}(x)}{\sum_k a_k p_{\mu_k}(x) p_{\mu_k}(x)} \, .$$

Note that $T_{\text{mean}}$ is not a Monge map since $\pi^*$ is not of the form $(\mathrm{Id}_d, T)_{\#}\mu$. In particular, $T_{\text{mean}\#}\mu$ is not equal to $\nu$ and $T_{\text{mean}}$ is not necessarily the gradient of a convex function. When using $MEW_2$ to obtain an assignment between two sets $\{x_i\}_i^M$ and $\{y_j\}_j^N$ of respectively $M$ and $N$ vectors of $\mathbb{R}^d$ and $\mathbb{R}^{d'}$, one can compute $T_{\text{mean}}(x_i)$ for each $x_i$, and then determine which $y_j$ is the closest of $T_{\text{mean}}(x_i)$ using a nearest-neighbor algorithm (Fix and Hodges, 1951).

### 4.2.3 Improving the $MGW_2$ method

Inspired by the $MEW_2$ method presented above, we propose in this section to improve the $MGW_2$ method by: (i) proposing an annealed scheme similarly to Algorithm 2 in order to reduce the chances of converging to sub-optimal local minima, (ii) designing a transportation plan for $MGW_2$ similarly to (9).

**Annealing scheme.** Since Problem ($MGW_2$) is non-convex, we are only guaranteed to converge towards a local minimum when solving it with a classic non-regularized GW solver (Peyré et al., 2016). Furthermore, the convergence towards a particular minimum depends strongly on the initialization of the coupling $\omega$. Since the discrete GW problem in $MGW_2$ is of very small scale and so not costly in itself, we propose, by anology with $MEW_2$, to use a similar annealing scheme as in Algorithm 2 to reduce the chance of converging to a sub-optimal local minimum. More precisely, this gives the following algorithm.

As previously, we set in our experiments $\alpha = 0.95$ and $\varepsilon_0 = 1$ as in Alvarez-Melis et al. (2019) and we observed that, in toy cases where we know what the global minimum is, that $N_{it} = 10$ seemed to be a sufficient number of iterations to prevent the algorithm from converging towards a sub-optimal minimum.

**Designing a transportation plan.** Still by analogy with $MEW_2$, one can design a transportation plan for $MGW_2$ by defining a matrix $P_{MGW_2} \in \mathbb{V}_{d'}(\mathbb{R}^d)$ and a vector $b_{MGW_2} \in \mathbb{R}^d$, and then replacing $\psi^* \circ T_{W_2}$ in (9) by $\psi_{MGW_2} \circ T_{W_2}$, where for all $x \in \mathbb{R}^d$, $\psi_{MGW_2}(x) = P_{MGW_2}^T(x - b_{MGW_2})$. More precisely, this can be done the following way. Given two GMMs $\mu = \sum_k a_k \mu_k$ and $\nu = \sum_l b_l \nu_l$ respectively in $GMM_K(\mathbb{R}^d)$ and $GMM_L(\mathbb{R}^{d'})$ and given the optimal discrete plan $\omega^*$ solution of Problem ($MGW_2$), one can define the matrix

---

**Algorithm 3** Annealed mixture Gromov-Wasserstein solver

---

**Require:** $\mu = \sum_k^K a_k \mu_k$, $\nu = \sum_l^L b_l \nu_l$, $\alpha \in (0,1)$, $\varepsilon_0$, $\omega^{\{0\}} = ab^T$
 1: $[C^x]_{k,i} \leftarrow W_2^2(\mu_k, \mu_i)$ for $k = 1, \ldots, K$, $i = 1, \ldots, K$
 2: $[C^y]_{l,j} \leftarrow W_2^2(\nu_l, \nu_j)$ for $l = 1, \ldots, L$, $j = 1, \ldots, L$
 3: **for** $n = 1, \ldots, N_{it}$ **do**
 4:      $\omega^{\{n\}} \leftarrow \varepsilon\text{-GW}(a, b, C^x, C^y, \varepsilon_{n-1}, \omega^{\{n-1\}})$             ▷ Solve a regularized GW problem.
 5:      $\varepsilon_n \leftarrow \alpha \varepsilon_{n-1}$             ▷ Annealing scheme.
 6: **end for**
 7: **return** SOLVE-GW$(a, b, C^x, C^y, \omega^{\{N_{it}\}})$           ▷ Solve the non-regularized GW problem.

---

$P_{MGW_2}$ as the solution of the following problem

$$\inf_{P \in \mathbb{V}_{d'}(\mathbb{R}^d)} \sum_{k,l} \omega_{k,l}^* W_2^2(\mu_k', P_\# \nu_l') \,, \tag{10}$$

where $\mu_k'$ and $\nu_l'$ are the Gaussian component of the centered GMMs $\bar{\mu}$ and $\bar{\nu}$, then we can set $b_{MGW_2} = \mathbb{E}_{X \sim \mu}[X] - P_{MGW_2} \mathbb{E}_{Y \sim \mu}[Y]$. As above, this problem can be solved numerically by performing a projected gradient descent on $P$, using either automatic differentiation or Lemma 10. This is also a non-convex optimization problem since $\mathbb{V}_{d'}(\mathbb{R}^d)$ is non-convex and so the solution given by the projected gradient descent depends on the initialization. We propose thus to initialize with the projection on the Stiefel manifold of the discrete cross-covariance matrix between the means of the Gaussian components, i.e.

$$P_{MGW_2}^{\{0\}} = \kappa_{\mathbb{V}_{d'}(\mathbb{R}^d)} \left( \sum_{k,l} \omega_{k,l}^* m_{0k} m_{1l}^T \right) \,.$$

Finally, using $P_{MGW_2}$ one can define a continous plan $\pi_{MGW_2}$ associated with the discrete optimal plan $\omega^*$ solution of the $MGW_2$ problem similarly to (9). We can therefore use $MGW_2$ to transport distributions, using as previously $T_{\text{mean}}$. We can also, as for $MEW_2$, use $MGW_2$ to obtain an assignment between two sets of points.

## 5 Experiments

In what follows, we use $MGW_2$ and $MEW_2$ to solve Gromov-Wasserstein related tasks on various datasets. More precisely, we apply first the two methods on simple toy low-dimensional GMMs. Then, we show that both methods can be used to solve relatively efficiently GW related tasks on real datasets in moderate to large scale settings involving sometimes several tens of thousands of points, both for evaluating distances between clouds of points and drawing correspondences between points. In all our experiments, we use the numerical solvers provided by the Python Optimal Transport (POT) package[5] (Flamary et al., 2021) that implements solvers for the non-regularized and regularized classic OT and GW problems. Code is available here[6].

### 5.1 Low dimensional GMMs

In Figure 4, we use again the example of Figure 3 and we derive an optimal transport plan for the $MGW_2$ problem as described in Section 4.2.3. We also show the plan obtained by solving the $EW_2$ problem. One can see that with both solutions, the global structure of the distribution is preserved in the sense that points that are closed to each other but in two different Gaussian components have been sent to points that are also close to each other but in different Gaussian components.

### 5.2 Distances between clouds of points

In this section, we illustrate the usability of our methods to assess distances between clouds of points. First, we reproduce an experiment originally conducted in Rustamov et al. (2013) and presented in Solomon et al.

---

[5]The package is accessible here: https://pythonot.github.io/.
[6]https://github.com/AntoineSalmona/MixtureGromovWasserstein

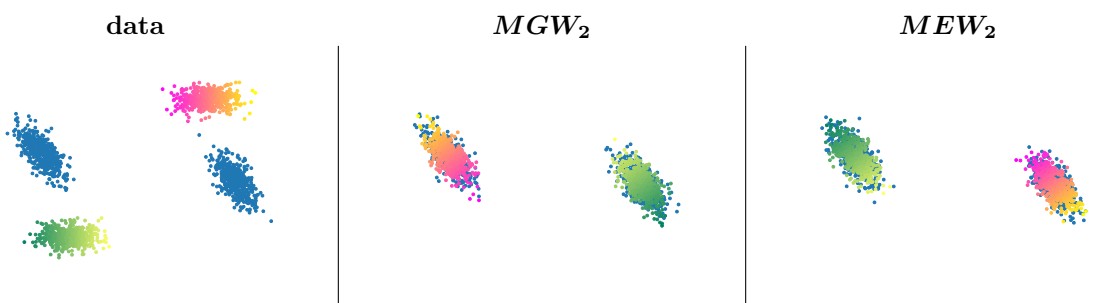

Figure 4: Left: two discrete distributions $\hat{\mu}$ (in gradient of colors) and $\hat{\nu}$ (in blue) that have been drawn from two GMMs. The colors have been added to $\hat{\mu}$ in order to visualize the couplings between $\hat{\mu}$ and $\hat{\nu}$. Middle: transport of $\hat{\mu}$ obtained by solving the $MGW_2$ problem, then deriving $P_{MGW_2} \in \mathbb{V}_2(\mathbb{R}^2)$ by solving Problem (10). Right: transport of $\hat{\mu}$ obtained by solving the $MEW_2$ problem.

(2016) with the use of entropic-regularized GW, that aims to recover the cyclical nature of a horse's gallop. Then, we perform a comparison between runtimes of $MGW_2$ and other methods existing in the literature that provide a GW-type distance between point clouds.

**Galloping horse sequence.** Here we repoduce the experiment of the galloping horse, that has been originally conducted in Rustamov et al. (2013) and presented in Solomon et al. (2016) with the use of entropic-regularized GW. In this experiment, we compute a matrix of pairwise distances (either for $MGW_2$ or $MEW_2$) between 45 meshes representing a galloping horse. Then, we conduct a Multi-Dimensional Scaling (MDS) (Borg and Groenen, 2005) - which roughly can be thought as a generalization of PCA - of the $45 \times 45$ matrix of pairwise distances between meshes, in order to plot each mesh as a 2-dimensional point. Figure 5 shows these 2-dimensional embeddings of the sequence. As observed in Solomon et al. (2016), the interesting part here is that these points are positioned in a cyclical fashion, which means that the original set of pairwise distances seem to respect the periodic aspect of the sequence (both for $MGW_2$ and $MEW_2$). Each mesh is composed of approximately 9000 vertices and the average time to compute one distance when using the POT implementation of the entropic-regularized GW solver is around 30 minutes which makes the computation of the full pairwise distance matrix impractical, as mentioned in Solomon et al. (2016). In constrast, when using our methods with GMMs with $K = 20$ components, it took us only approximately 10 minutes to compute the full distance matrix using $MGW_2$, and around one hour using $MEW_2$, these times including the fitting with EM of all the GMMs.

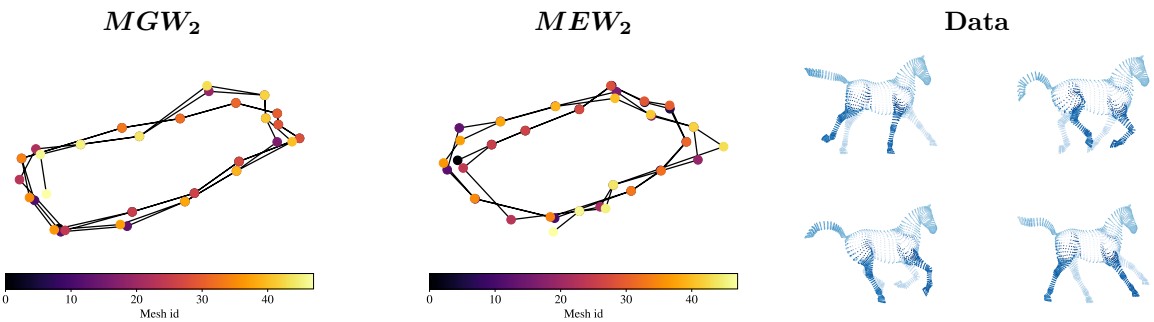

Figure 5: MDS on the galloping horse animation using the $MGW_2$ distance (left), and the $MEW_2$ distance (middle). Each point corresponds to a given mesh and the meshes are colored in function of their number in the sequence. Right: 4 examples among the 45 meshes that composes the sequence. The computations of both distances have been done by first fitting GMMs with 20 components on each mesh independently.

**Local minima.** To highlight the importance of using an annealing scheme when deriving $MGW_2$ or $MEW_2$, we have reconducted the previous experiment but this time without the annealing schemes described in Algorithm 3 and Algorithm 2. In Figure 6, we plot the evolutions of the values of $MGW_2^2$ and $MEW_2^2$ between one given fixed mesh and all the others. In both cases, the annealing scheme seems to be useful to prevent the solver to converge towards sub-optimal mininima. However, if the $MGW_2$ solver seems to often converge to the same optimum regardless the use of the annealing scheme, this is not the case of $MEW_2$ which, without the annealing initialization procedure (Algorithm 2), converges most of the time to a sub-optimal minimum, so much that the periodical aspect doesn't even appear in that case. This experiment also emphasizes the fact that when solving a GW problem with classic non-regularized or entropic solvers from Peyré et al. (2016), we are not at all guaranteed to converge towards a global minimum and, more critically, we have in general no ways to know if the solution we converged to is actually optimal or sub-optimal.

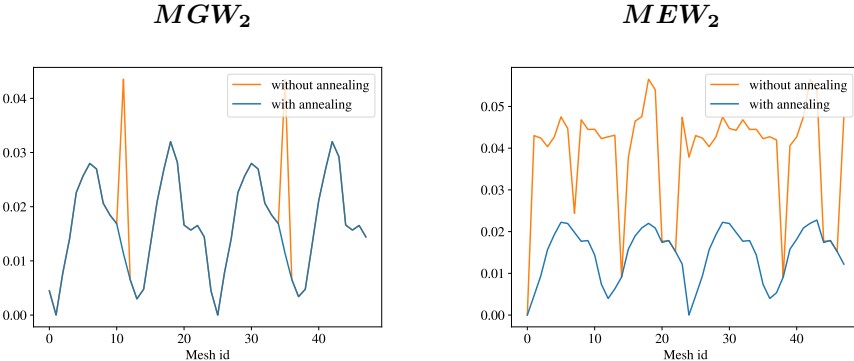

Figure 6: Left: Evolution of $MGW_2^2$ between the second mesh and all the others, using an annealing scheme (Algorithm 3) in blue, and without the annealing scheme in orange. Right: Evolution of $MEW_2^2$ between the first mesh and all the others, with the annealing initialization procedure (Algorithm 2) in blue, and without in orange. The computation of both distances has been done by first fitting GMMs with 20 components on each mesh independently.

**Runtimes comparison.** We perform a comparison between runtimes of $MGW_2$, sliced GW (SGW) (Vayer et al., 2019b), low-rank GW (lrGW)(Scetbon et al., 2022), minibatch GW (mbGW) (Fatras et al., 2021), entropic-regularized GW (erGW) (Peyré et al., 2016) and quantized GW (qGW) (Chowdhury et al., 2021) between two 2D random discrete distributions with varying number of points from $n = 10^3$ to $n = 10^6$. We use the codes provided by the authors on their dedicated Github repositories. Note that $MEW_2$ is not included in this comparison as we observed in the previous experiment that this latter method was significantly slower than $MGW_2$. For $MGW_2$, we use GMMs with respectively $K = \{10, 20, 50\}$ components. For SGW, we use the implementation on CPU with $L = \{50, 200\}$ projections. lrGW has a parameter $r$ corresponding to the rank of the coupling matrix. We choose here respectively $r = n/100$ (this choice is advised by the authors of (Scetbon et al., 2022) for lrGW to be a good approximation of erGW) and $r = 100$ (which yields a linear computational time). For mbGW, we use batches of size $m = 50$ with $k = n/10$ batches (these values are advised by the authors of (Fatras et al., 2021)). For erGW, we use two different implementations of the method, the first one from POT and the second from Scetbon et al. (2022)[7], both with regularization parameter $\varepsilon = 0.1$. Finally, for qGW, we use a proportion $p = 0.1$ of the points as partition block representatives and then we take a Voronoi partition with respect to these representatives. Note that this latter method only provides a coupling but we reinject it in the GW objective. Results can be found in Figure 7. We can observe that $MGW_2$ has similar runtimes as SGW (CPU version) and seems even a bit faster in large scale settings. Several algorithms fail to converge when the number of points is too large. The limits we observed are: $10^4$ for both implementations of erGW, $2 \times 10^4$ for lrGW with $r = n/100$, and $4 \times 10^4$ for lrGW with $r = 100$. In the same way, considering our computational ressources, using qGW to compute a distance between the two point clouds with more than $3 \times 10^4$ points was impossible because the two full pair-to-pair distance matrices are becoming too heavy in terms of memory. Note that it is still possible to

---

[7]https://github.com/meyerscetbon/LinearGromov/blob/main/FastGromovWass.py

compute a coupling using qGW afterwards, see (Chowdhury et al., 2021) for details, but it is no more possible to evaluate the GW objective, which necessarily requires to access the full pair-to-pair distance matrices.

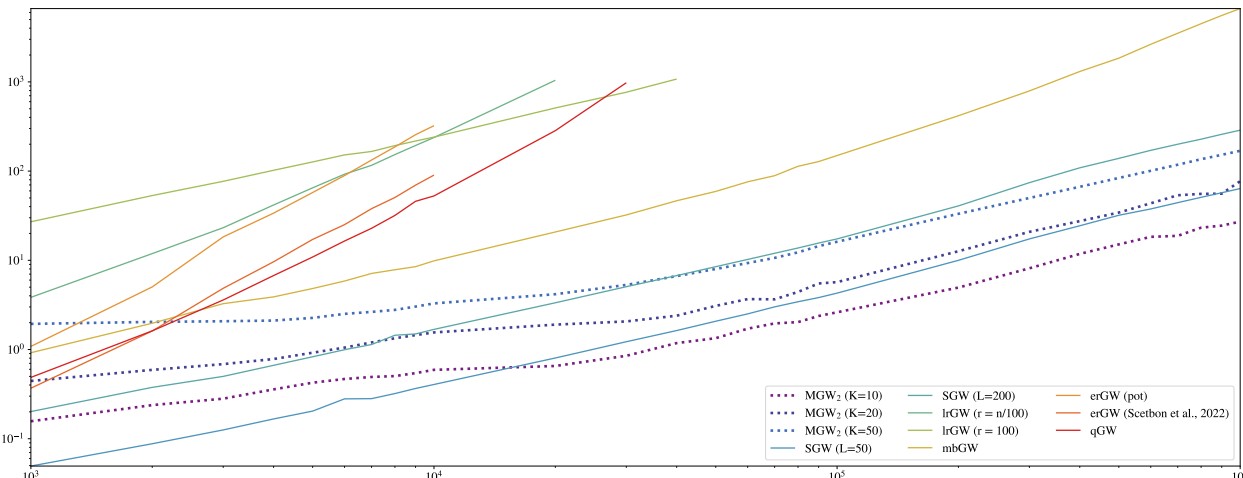

Figure 7: Runtimes comparison between $MGW_2$, SGW (Vayer et al., 2019b) (CPU), lrGW (Scetbon et al., 2022), mbGW (Fatras et al., 2021), erGW (Peyré et al., 2016) and qGW (Chowdhury et al., 2021) between two 2D random discrete distributions with varying number of points from $10^3$ to $10^6$ in log-log scale. The time includes the computation of the pair-to-pair distance matrices and the fitting of the GMMs for $MGW_2$ (using scikit-learn).

### 5.3 Drawing correspondences between points

In this section, we illustrate the usability of our methods to establish correspondences between clouds of points on two shape matching applications.

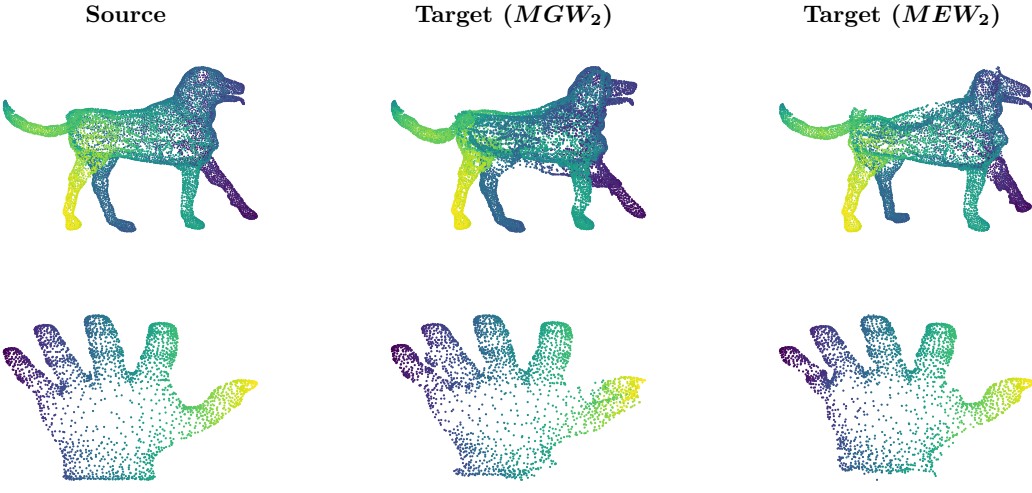

Figure 8: Shape matching between shapes and their distorted versions. We plot the output of the $T_{\text{mean}}$ transport map applied to the source shape on the left for respectively $MGW_2$ (middle) and $MEW_2$ (right). GMMs with 20 components have been fitted independently on the source shape and on its noisy version. Colors have been added to visualize where the points have been transported.

**Quality of the transport map.** We reproduce here an experiment from Chowdhury et al. (2021). The goal is to match 3D meshes from the CAPOD dataset (Papadakis, 2014) with copies of themselves whose

vertices are permuted and perturbed randomly. To do so, we fit a GMM on each mesh as well as on its noisy version, then we derive a discrete coupling $\omega$ between the Gaussian components using $MGW_2$ or $MEW_2$. Finally, we transport the points using $T_{\text{mean}}$ (see Section 4.2.2). Two examples of matching using GMMs with 20 components can be found in Figure 8. Observe that both methods $MGW_2$ and $MEW_2$ seem to be able to recover relatively well the correct matching of the points on these examples. To complete this experiment with quantitative results, we compute, as in Chowdhury et al. (2021), the distortion at each point $x$ as the distance from its ground truth copy $x$ and its matched point $y$ with a given coupling $\pi$. The distortion score of the coupling $\pi$ is then the mean squared distortion. We report class average distortion scores for $MGW_2$ and $MEW_2$ with GMMs with $K = \{10, 20, 50\}$ components as well as computation times in Table 1. Since the points at the output of $T_{\text{mean}}$ are not exactly corresponding to the points in the noisy version of the shapes, we introduce an additional nearest neighbors step in order to reproject the points onto the noisy shape. We also report in Table 1 class average distortion scores and computation times for qGW using a proportion $p = \{0.01, 0.1, 0.2, 0.5\}$ of the points as partition block representatives and using a Voronoi partition with respect to these representatives. In terms of distortions scores, we observe that $MGW_2$ yields similar results that qGW for most classes. We remark that the results we obtained on qGW are consistent with the ones reported in (Chowdhury et al., 2021, Table 1). Note that Chowdhury et al. (2021) have shown that qGW significantly outperforms MREC (Blumberg et al., 2020) and mbGW (Fatras et al., 2021) on this task (which is why we don't include them in the comparison). In terms of running time, note that the situation is a little bit different from Figure 7 since the $MGW_2$ method includes here an additional step of deriving a coupling between the points given the coupling between the Gaussian components whereas the qGW method doesn't require to assess the GW objective after computing the coupling anymore. We observe that both methods $MGW_2$ and $MEW_2$ are significantly slower than qGW in that setting. Yet the gap of running time between qGW and our methods seems to reduce as the number of points increases. Indeed, using $MGW_2$ or $MEW_2$ seems to largely decorrelate computation time from the number of points (since the number of Gaussian component stays fixed), without observable deterioration in terms of registration accuracy. These methods therefore make a lot of sense as the number of points increases.

| Method | Param | Humans 1926 | Planes 2144 | Spiders 2664 | Cars 5220 | Dogs 8937 | Trees 10433 | Vases 15828 |
|---|---|---|---|---|---|---|---|---|
| $MGW_2$ | 10 | **0.04** (17.5) | 0.40 (17.6) | 0.03 (17.7) | **0.12** (17.8) | 0.13 (18.8) | 0.10 (18.8) | 0.34 (19.1) |
| | 20 | 0.15 (69.4) | 0.36 (69.5) | 0.04 (69.9) | 0.17 (70.8) | 0.20 (71.9) | 0.10 (71.9) | 0.28 (73.1) |
| | 50 | 0.18 (431) | 0.10 (428) | **0.007** (431) | 0.12 (435) | 0.20 (437) | **0.04** (438) | 0.20 (441) |
| $MEW_2$ | 10 | 0.09 (17.1) | 0.37 (22.9) | 0.02 (16.3) | 0.23 (17.6) | 0.20 (24.5) | 0.11 (18.3) | 0.29 (23.2) |
| | 20 | 0.21 (77.2) | 0.39 (66.6) | 0.02 (64.1) | 0.25 (78.0) | 0.20 (85.6) | 0.13 (67.1) | 0.30 (76.8) |
| | 50 | 0.16 (555) | 0.17 (421) | 0.009 (397) | 0.20 (423) | 0.21 (462) | 0.08 (465) | 0.19 (486) |
| $qGW$ | 0.01 | 0.25 (0.59) | 0.46 (0.78) | 0.05 (1.08) | 0.24 (3.88) | 0.28 (11.3) | 0.13 (17.4) | 0.28 (32.4) |
| | 0.1 | 0.16 (1.04) | 0.10 (1.33) | 0.02 (1.84) | 0.21 (5.80) | 0.02 (16.5) | 0.05 (26.8) | **0.18** (54.9) |
| | 0.2 | 0.11 (1.65) | 0.08 (2.12) | 0.01 (2.86) | 0.15 (9.25) | 0.008 (28.2) | **0.04** (53.6) | 0.21 (123) |
| | 0.5 | 0.10 (4.39) | **0.07** (5.77) | **0.007** (7.73) | 0.16 (34.9) | **0.007** (104) | 0.15 (165) | 0.22 (418) |

Table 1: Distortion scores (lower is better) and runtimes (in parentheses) for $MGW_2$, $MEW_2$, and qGW. The average number of points in each shape class is provided under the shape class name. Results are listed for several parameter choices of each method. Results have been averaged on 10 runs of the experiment.

**Matching human shaped meshes.** To demonstrate the usability of our methods in larger scale settings, we use the SHREC'19 dataset (Melzi et al., 2019) that contains human shaped meshes that can sometimes be composed of more than 300000 vertices. Our goal is to draw correspondences between the shapes using only the information of the vertices (the dataset also includes edges). To do so, we first fit independently GMMs with 20 components on each mesh and we derive directly couplings at the scale of the Gaussian components that represent the different parts of the bodies. In such large scale settings, the main bottleneck of the methods in terms of computational time is clearly the fitting of the GMMs that can take at worst 2 minutes for the meshes composed of the highest number of vertices. The results are displayed on Figure 9. Observe that in most cases, both methods seem to be able to match correctly the colored parts. Yet in the last row, $MEW_2$ matches a leg at the left in red to an arm at the right. This probably implies that the method has been trapped in a local minimum despite the annealing initialization procedure. Finally, note that we presented here cases where the methods performed relatively well, but there are cases where $MGW_2$

or $MEW_2$ fail to find correct correspondences and exhibit behaviors similar to $MEW_2$ in the last row, which suggests that the methods converge sometimes to sub-optimal minima despite the annealing schemes.

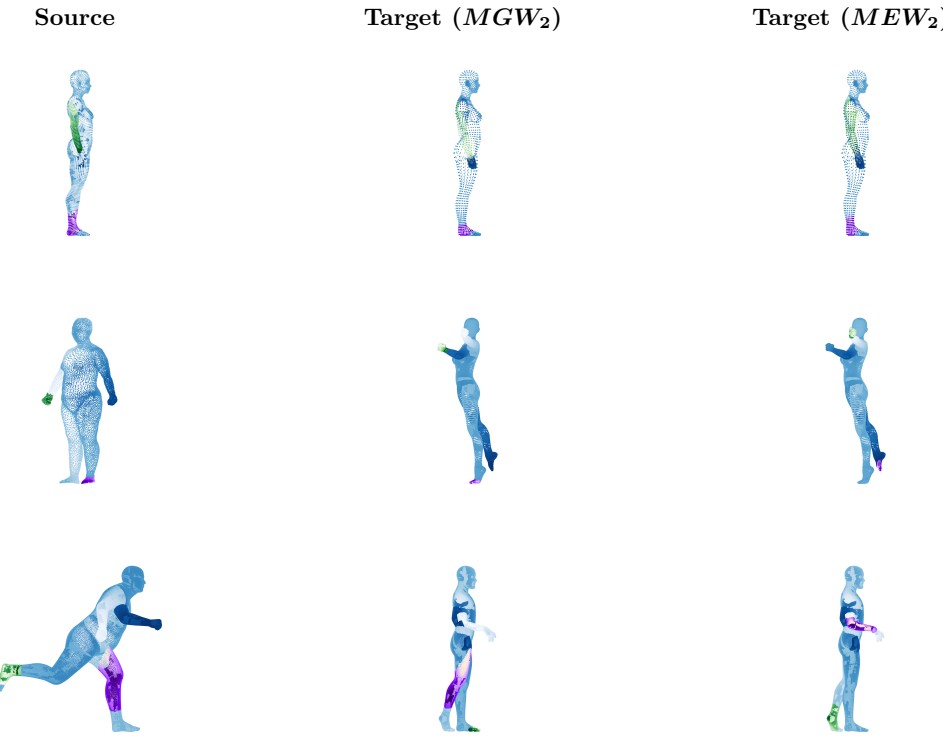

**Source**   **Target ($MGW_2$)**   **Target ($MEW_2$)**

Figure 9: Shape matching between human-shaped meshes using $MGW_2$ (middle) and $MEW_2$ (right). Each shape on the left column is matched with the shapes on the same row. GMMs with 20 components have been fitted independently on each shape and the points colored in green and purple correspond to Gaussian components that are matched together when solving $MGW_2$ or $MEW_2$. From left to right and top to bottom, the meshes are composed respectively of 84912, 30300, 75000, 273624, 360678, and 360357 vertices.

## 5.4 Application to hyperspectral image color transfer

The goal here is to reproduce the experiment of color transfer conducted in Delon and Desolneux (2020), but this time using a hyperspectral image, i.e an image with more than 3 color channels. More precisely, we aim to create an RGB image from an hyperspectral image $u$ using the colors of another RGB image $v$. To do so, we consider images as empirical distributions in the color spaces and we solve a Gromov-Wasserstein problem between the distributions $\hat{\mu} = \frac{1}{M}\sum_k^M \delta_{u_k}$ and $\hat{\nu} = \frac{1}{N}\sum_l^N \delta_{v_l}$, where $M$ and $N$ are the number of pixels in respectively the hyperspectral image and the RGB image we use as color palette, and $\{u_k\}_k^M$ and $\{v_l\}_l^N$ are the values at each pixel, i.e for here all $l$, $v_l \in \mathbb{R}^3$ and for all $k$, $u_k \in \mathbb{R}^d$ with $d > 3$. We thus fit two GMMs $\mu$ and $\nu$ on respectively $\hat{\mu}$ and $\hat{\nu}$ and we use $MGW_2$ or $MEW_2$ to derive a mapping $T_{\text{mean}} \colon \mathbb{R}^d \to \mathbb{R}^3$, as described in Section 4.2.2. We apply this process to a hyperspectral image of $512 \times 512$ pixels with 15 channels that are displayed in Figure 10 top left. We use as color palettes two paintings by Gauguin and Renoir, displayed in Figure 10 top right, that are respectively *Manhana no atua* (top) and *Le déjeuner des canotiers* (bottom). These two images are composed of $1024 \times 768$ pixels. The resulting images $T_{\text{mean}}(u)$ are displayed in Figure 10 bottom (Gauguin at the left and Renoir at the right). For this experiment, we observed that setting the number of Gaussian components to $K = 15$ was a good compromise between capturing the complexity of the color distributions and obtaining a relatively regular mapping $T_{\text{mean}}$. This experiment shows that $MGW_2$ and $MEW_2$ can be used in large scale settings: observe indeed that the color distributions $\hat{\mu}$ and $\hat{\nu}$ are composed respectively of approximatively 300000 and 800000 points, which makes the problem intractable with entropic-GW solvers such as Peyré et al. (2016) or Solomon et al. (2016).

In term of computation time, the fitting of the two GMMs for the hyperspectral image takes aproximatively one minute against 20 seconds for the GMM for the RGB image. The projected gradient descent becomes rather slow in that setting, which makes it preferable to few updates of $P$ at each step of Algorithm 1 for the computation of $MEW_2$. Finally, for both methods, it takes around 2 minutes to compute the whole RGB image $T_{\text{mean}}(u)$.

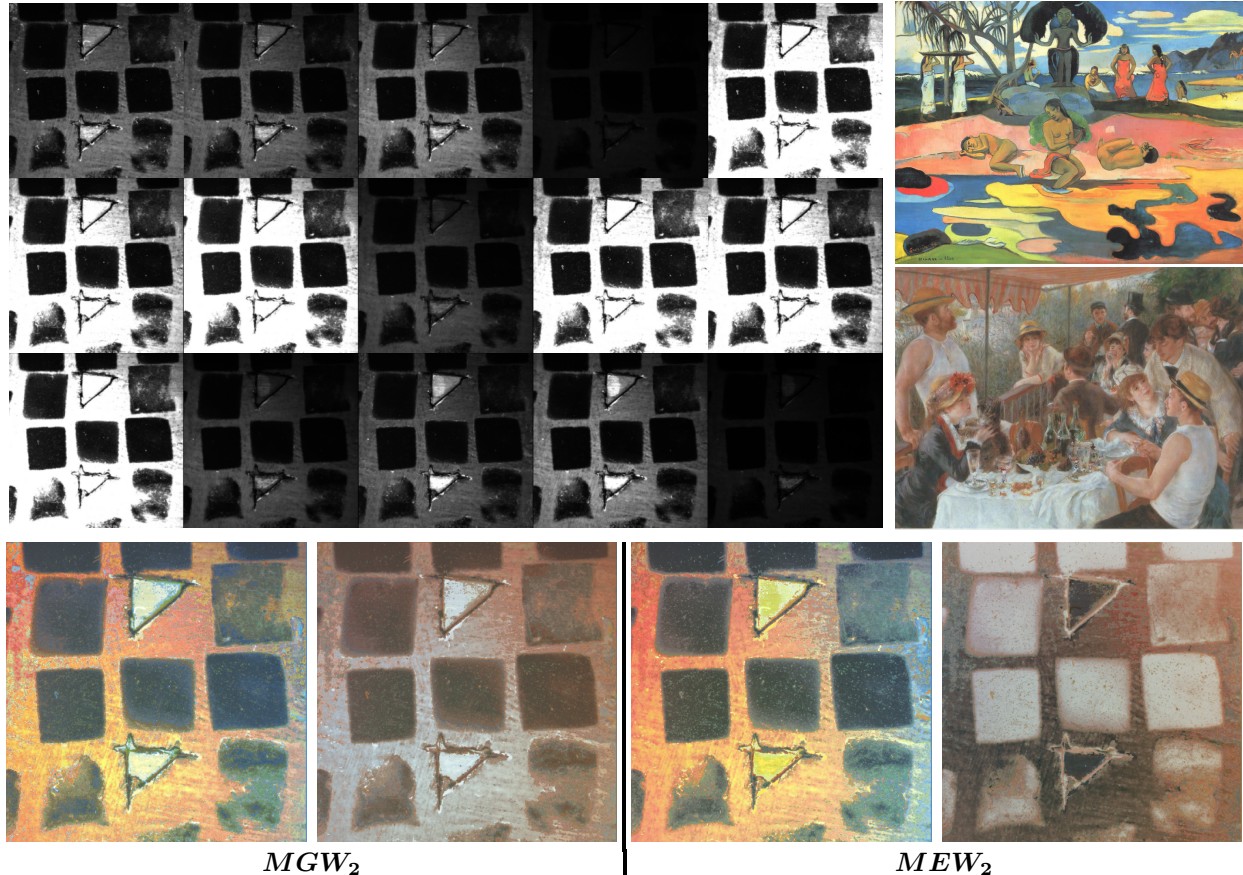

Figure 10: Color transfers between a hyperspectral image with 15 channels (top left) and two paintings by Gauguin and Renoir (top right, middle right). Bottom line: the obtained RGB images using $MGW_2$ and $MEW_2$. For this experiment, we used GMMs with 15 components. Image taken by Francesca Ramacciotti (Alma Mater Studiorum - University of Bologna) and Laure Cazals (supported by the European Commission in the framework of the GoGreen project (GA no. 101060768)).

## 6 Conclusion and perspectives

In this paper, we have introduced two new OT distances on the set of Gaussian mixture models, $MGW_2$ and $MEW_2$, and we have shown that they both can be used to solve relatively efficiently Gromov-Wasserstein related problems on Euclidean spaces, especially in moderate-to-large scale settings involving several tens of thousands of points. These OT distances are also by design particularly suited to settings where there already exists a kind of clustering structure in the data. This being said, if $MEW_2$ remains an efficient alternative to the entropic GW solvers proposed by Peyré et al. (2016) and Solomon et al. (2016), we observed that the method was actually slower and perhaps harder to tune than $MGW_2$ for a slighty lower quality of results, and so we believe that $MGW_2$ is a better choice in practice. This latter distance is part of the family of Gromov-Wasserstein type OT distances that reduce the size of the GW problem, which also includes notably qGW (Chowdhury et al., 2021), MREC (Blumberg et al., 2020) and scalable GW (Xu et al., 2019). To the

best of our knowledge, no such method specifically based on Gaussian mixture clustering had already been proposed in the literature. Furthermore, our method differs from these three other approaches in the fact that we only need here to solve numerically one single GW problem at the scale of the clusters, using not only the centroid position information but also order 2 statistics. Note in particular that our method has strong similarities with qGW, which uses Voronoi quantizations of the spaces instead of GMMs However, an important difference lies in the fact that the $MGW_2$ method primarily provides a distance between clouds of points before providing a heuristic coupling between them, while the qGW method directly derives a heuristic coupling which can then be reinjected into the objective function to derive a distance. For both methods, the optional additional step of computing a coupling for $MGW_2$ and computing a distance for qGW increases the computation time. Consequently, $MGW_2$ seems to be a more appropriate choice than qGW for tasks that require only a distance between clouds of points, see the computation times of Figure 7, while qGW seems to be a more appropriate choice for tasks that require only a coupling between points. Still, we have shown in our experiments that $MGW_2$ can also provide couplings, with equivalent performance to qGW in terms of accuracy, although it is significantly slower in the small-to-medium scale setting of Table 1. However, we believe that one advantage of $MGW_2$ over qGW for deriving couplings is that $MGW_2$ appears to better decorrelate the number of clusters needed to achieve good accuracy from the number of points, which may become an important feature in larger scale settings.

**Perspectives for future work** $MGW_2$ could be easily extended to other type of mixtures as soon as we have an identifiability property between the mixtures and the probability distributions on the space of the distributions that compose the mixtures. If in the Euclidean setting GMMs seem to be versatile enough to represent large classes of concrete and applied problems, an interesting extension of our work could be to consider mixture of distributions on non-Euclidean spaces.

Computationally speaking, the main bottleneck of the method probably comes from the fitting of the GMMs with the Expectation-Maximization (EM) algorithm (Dempster et al., 1977) which can become relatively costly in large scale settings or as soon as the dimension increases. If the EM algorithm remains invariably the classical algorithm for learning GMMs, some recent approaches (Hosseini and Sra, 2020; Sembach et al., 2022; Pasande et al., 2022) have proposed alternative algorithms that seem to outperform it. These approaches are based on Riemannian stochastic optimization, leveraging the rich Riemannian structure of the set of positive definite matrices. Another interesting alternative that has been shown to outperform the EM algorithm has been proposed by (Kolouri et al., 2018) and is based on the minimization of the sliced-Wasserstein distance. Integrating this in our method could result thus in an approach fully-based on optimal transport.

Another possible limitation of our work lies in the fact that the $MGW_2$ solver converges sometimes to sub-optimal local minima. If the annealed procedure introduced in Section 4.2.3 seems to reduce this issue, we generally have no guarantee that the solution we have converged to is optimal. This is not specific to our method and comes from the gradient descent structure of the classic GW solvers. Still, when solving the GW problem between GMMs rather than solving it directly between the points, it is likely that we increase the probability of converging towards a sub-optimal local minimum because we inevitably introduce symmetries by simplifying the problem and so we probably increase in the mean time the number of local minima in the GW objective. In the Euclidean setting, the recent work of Ryner et al. (2023) proposes an algorithm for solving the GW problem that is guaranteed to converge toward a global minimum, leveraging the low-rank structure of the cost matrices when the cost functions are the squared Euclidean distances. A future perspective of work could be therefore to study if a similar idea could be applied for solving the $MGW_2$ problem.

### Acknowledgements

This research was funded, in part, by the Agence nationale de la recherche (ANR), through the SOCOT project (ANR-23-CE40-0017), and the PEPR PDE-AI project (ANR-23-PEIA-0004).

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

## Organization of the supplementary

The supplementary is organized as follows. First, in Appendix A, we show six technical results that will be used throughout the proofs of the paper. In Appendix B, we give the full proofs of the technical results of the paper. Finally, in Appendix C, we give more details on the difference between $EW_2$ and the OT distance introduced by Cai and Lim (2022).

## A  Technical lemmas

Before turning to the proofs of the theoretical results, we state here six technical lemmas that will be used throughout the proofs of the results of the paper.

### A.1  A property of couplings between measures living in different dimensions

First we start by recalling the following result (Delon et al., 2022b, Lemma 3.3).

**Lemma A1 (Delon et al., 2022b).** *Let $\mu \in \mathcal{W}_2(\mathbb{R}^d)$ and $\nu \in \mathcal{W}_2(\mathbb{R}^{d'})$ with $d$ not necessarily greater than $d'$, and let $T \colon \mathbb{R}^{d'} \to \mathbb{R}^d$ be a measurable map. Then $\pi' \in \Pi(\mu, T_{\#}\nu)$ if and only if there is some $\pi \in \Pi(\mu, \nu)$ such that $\pi' = (\mathrm{Id}_d, T)_{\#}\pi$. In particular, if there exist $a, b \geq 0$ such that $\|T(y)\| \leq a + b\|y\|$ for all $y \in \mathbb{R}^{d'}$, then*

$$\inf_{\pi \in \Pi(\mu,\nu)} \int_{\mathbb{R}^d \times \mathbb{R}^{d'}} \|x - T(y)\|^2 \mathrm{d}\pi(x,y) = \inf_{\pi \in \Pi(\mu, T_{\#}\nu)} \int_{\mathbb{R}^d \times \mathbb{R}^d} \|x - z\|^2 \mathrm{d}\pi(x,z) \ .$$

### A.2  Isometries in Euclidean spaces

We show the following result, that states that any isometry $T \colon \mathbb{R}^{d'} \to \mathbb{R}^d$ for the Euclidean norm is affine and of the form, for all $y \in \mathbb{R}^{d'}$, $T(y) = Py + b$, where $b \in \mathbb{R}^d$ and $P$ is in the Stiefel manifold $\mathbb{V}_{d'}(\mathbb{R}^d)$.

**Lemma A2.** *Suppose $d \geq d'$. Then $\phi \colon \mathbb{R}^{d'} \to \mathbb{R}^d$ is an isometry for the Euclidean norm if and only if there exist $P \in \mathbb{V}_{d'}(\mathbb{R}^d)$ and $b \in \mathbb{R}^d$ such that for all $y \in \mathbb{R}^{d'}$, $\phi$ is of the form*

$$\phi(y) = Py + b \ .$$

*Proof.* First observe that for $P \in \mathbb{V}_{d'}(\mathbb{R}^d)$ and $b \in \mathbb{R}^d$, $y \mapsto Py + b$ is an isometry since we have, for any $y$ and $y'$ in $\mathbb{R}^{d'}$

$$\|Py + b - Py' - b\|^2 = \|P(y - y')\|^2 = (y - y')^T P^T P(y - y') = (y - y')^T(y - y') = \|y - y'\|^2 \ .$$

The converse is a consequence of the Mazur–Ulam theorem (Mazur and Ulam, 1932) that states - in the version of Baker (1971) - that an isometry from a real normed space to a *strictly convex* normed space, i.e. a normed space where the unit ball is a stricly convex set, is necessarily affine. Since it is easy to show that the unit ball $\{x \in \mathbb{R}^d : \|x\| \leq 1\}$ is a strictly convex set, we get that for all $x \in \mathbb{R}^{d'}$, $\phi$ is of the form $y \mapsto Py + b$ with $P$ being a matrix of size $d \times d'$, and $b \in \mathbb{R}^d$. Moreover we have for all $y, y' \in \mathbb{R}^{d'}$

$$\|\phi(y) - \phi(y')\|^2 = \|Py - Py'\|^2 = \|P(y - y')\|^2 = (y - y')^T P^T P(y - y') \ .$$

Since $\phi$ is an isometry, it follows that $\|y - y'\|^2 = (y - y')^T P^T P(y - y')$ and so $P^T P = \mathrm{Id}_{d'}$, which concludes the proof. $\square$

### A.3  Centering of measures

**Lemma A3.** *Let $\mu \in \mathcal{W}_2(\mathbb{R}^d)$ and $\nu \in \mathcal{W}_2(\mathbb{R}^{d'})$ with $d$ not necessarily greater than $d'$. Let $\bar{\mu}$ and $\bar{\nu}$ denote the centered measures associated to $\mu$ and $\nu$ and let $\mathfrak{P}$ be any subset of matrices of size $d \times d'$. Then,*

$$\inf_{\pi \in \Pi(\mu,\nu)} \inf_{P \in \mathfrak{P}, \ b \in \mathbb{R}^d} \int_{\mathbb{R}^d \times \mathbb{R}^{d'}} \|x - Py - b\|^2 \mathrm{d}\pi(x,y) = \inf_{\pi \in \Pi(\bar{\mu},\bar{\nu})} \inf_{P \in \mathfrak{P}} \int_{\mathbb{R}^d \times \mathbb{R}^{d'}} \|x - Py\|^2 \mathrm{d}\pi(x,y) \ .$$

*Proof.* Denoting $m_0 = \mathbb{E}_{X \sim \mu}[X]$, $m_1 = \mathbb{E}_{Y \sim \nu}[Y]$, $\tilde{x} = x - m_0$, and $\tilde{y} = y - m_1$, we have for any $\pi \in \Pi(\mu, \nu)$,

$$\int_{\mathbb{R}^d \times \mathbb{R}^{d'}} \|x - Py - b\|^2 \mathrm{d}\pi(x, y) = \int_{\mathbb{R}^d \times \mathbb{R}^{d'}} \|\tilde{x} - P\tilde{y} - b + m_0 - Pm_1\|^2 \mathrm{d}\pi(x, y)$$

$$= \|m_0 - b - Pm_1\|^2 + \int_{\mathbb{R}^d \times \mathbb{R}^{d'}} \|\tilde{x} - P\tilde{y}\|^2 \mathrm{d}\pi(x, y) ,$$

since $\int \langle \tilde{x} - P\tilde{y}, m_0 - b - Pm_1 \rangle \mathrm{d}\pi(x, y) = 0$. Thus it follows,

$$\inf_{\pi \in \Pi(\mu, \nu)} \inf_{P \in \mathfrak{P}, \ b \in \mathbb{R}^d} \int_{\mathbb{R}^d \times \mathbb{R}^{d'}} \|x - Py - b\|^2 \mathrm{d}\pi(x, y)$$

$$= \inf_{P \in \mathfrak{P}} \left( \inf_{b \in \mathbb{R}^d} \|m_0 - Pm_1 - b\|^2 + \inf_{\pi \in \Pi(\bar{\mu}, \bar{\nu})} \int_{\mathbb{R}^d \times \mathbb{R}^{d'}} \|x - Py\|^2 \mathrm{d}\pi(x, y) \right) .$$

Observe now that for any $P \in \mathfrak{P}$, $\|m_0 - Pm_1 - b\|^2 = 0$ if $b = m_0 - Pm_1$, which concludes the proof. $\square$

## A.4 A matrix linear program

**Lemma A4.** *Let $K$ be a matrix of size $d \times d'$ with Singular Value Decomposition (SVD) $K = U_K \Sigma_K V_K^T$ and let $\mathfrak{P}$ be any compact set of matrices of size $d \times d'$. Then,*

$$\sup_{P \in \mathfrak{P}} \mathrm{tr}(P^T K) = \max_{P \in \mathfrak{P}} \mathrm{tr}(\Sigma_P^T \Sigma_K) ,$$

*where $\Sigma_P = \mathrm{diag}^{[d, d']}(\boldsymbol{\sigma}(P))$ with $\boldsymbol{\sigma}(P) \in \mathbb{R}_+^{d'}$ denoting the vector of singular values of $P$. Furthermore the supremum is achieved at $P$ of the form,*

$$P = U_K \Sigma_P V_K^T .$$

*Proof.* Note that this lemma can be proven with a proof similar to the one of Alvarez-Melis et al. (2019, Lemma 4.2), using the min-max theorem for singular values. Here we offer an alternative proof based on Lagragian analysis. First observe that the supremum is achieved as a direct consequence of the Weierstrass theorem because $\mathfrak{P}$ is compact and the mapping $P \mapsto \mathrm{tr}(P^T K)$ is continuous. For a given $P \in \mathfrak{P}$, let $U_P \Sigma_P V_P^T$ be the SVD of $P$. The problem can be rewritten as

$$\max_{P \in \mathfrak{P}} \mathrm{tr}(V_P \Sigma_P^T U_P^T U_K \Sigma_K V_K^T) .$$

Now, let us denote $U = U_P^T U_K$ and $V = V_P^T V_K$. Observe that $U$ is in $\mathbb{O}(\mathbb{R}^d)$ and $V$ is in $\mathbb{O}(\mathbb{R}^{d'})$. Using the cyclical permutation of the trace operator, the problem becomes

$$\max_{P \in \mathfrak{P}} \mathrm{tr}(\Sigma_P^T U \Sigma_K V^T) .$$

Now, for a given fixed $\Sigma_P$, we determine which $U$ and $V$ maximize $\mathrm{tr}(\Sigma_P^T U \Sigma_K V^T)$. This problem reads as

$$\max_{U \in \mathbb{O}(\mathbb{R}^d), \ V \in \mathbb{O}(\mathbb{R}^{d'})} \mathrm{tr}(\Sigma_P^T U \Sigma_K V^T) .$$

The Lagrangian of this problem reads as

$$\mathcal{L}(U, V, C_0, C_1) = -\mathrm{tr}(\Sigma_P^T U \Sigma_K V^T) + \mathrm{tr}(C_0(U^T U - \mathrm{Id}_d)) + \mathrm{tr}(C_1(V^T V - \mathrm{Id}_{d'})) ,$$

where $C_0 \in \mathbb{S}^d$ and $C_1 \in \mathbb{S}^{d'}$ are the Lagrange multipliers respectively associated with the constraints $U \in \mathbb{O}(\mathbb{R}^d)$ and $V \in \mathbb{O}(\mathbb{R}^{d'})$. The first order condition gives

$$\begin{cases} \Sigma_P V \Sigma_K^T = 2UC_0 \\ \Sigma_P^T U \Sigma_K = 2VC_1 , \end{cases}$$

or equivalently

$$\begin{cases} U^T \Sigma_P V \Sigma_K^T = 2C_0 \\ \Sigma_P^T U \Sigma_K V^T = 2V C_1 V^T \ . \end{cases}$$

Since $C_0$ and $C_1$ are symmetric matrices (because they are associated with symmetric constraints), we get that both left-hand terms are symmetric. This gives the following conditions

$$\begin{cases} U^T \Sigma_P V \Sigma_K^T = \Sigma_K V^T \Sigma_P^T U \\ \Sigma_P^T U \Sigma_K V^T = V \Sigma_K^T U^T \Sigma_P \ . \end{cases}$$

Now, observe that when multiplying the first condition at right by $U^T \Sigma_P$ and multiplying the second condition at left by $\Sigma_K V^T$, we get by combining the two conditions

$$\begin{cases} U \Sigma_K V^T \Sigma_P^T \Sigma_P = \Sigma_P \Sigma_P^T U \Sigma_K V^T \\ U^T \Sigma_P V \Sigma_K^T \Sigma_K = \Sigma_K \Sigma_K^T U^T \Sigma_P V^T \ , \end{cases}$$

or equivalently,

$$\begin{cases} U \Sigma_K V^T D_P = D_P^{[d]} U \Sigma_K V^T \\ U^T \Sigma_P V D_K = D_K^{[d]} U^T \Sigma_P V^T \ , \end{cases}$$

where $D_P = \mathrm{diag}(\boldsymbol{\sigma}(P))$ and $D_K = \mathrm{diag}(\boldsymbol{\sigma}(K))$. Multiplying the first condition at left by $V \Sigma_K^T U^T$ and the second condition at right by $V \Sigma_P^T U$, this yields to

$$\begin{cases} V D_K V^T D_P = V \Sigma U^T D_P^{[d]} U \Sigma_K V^T \\ D_K^{[d]} U^T D_P^{[d]} U = U^T \Sigma_P V D_K V \Sigma_P^T U \ . \end{cases}$$

It follows that $V D_K V^T D_P$ and $D_K^{[d]} U^T D_P^{[d]} U$ are symmetric matrices and so $V D_K V^T$ commutes with $D_P$ and $U^T D_P^{[d]} U$ commutes with $D_K^{[d]}$. Thus we can deduce that $U$ and $V$ are permutation matrices. Since the singular values are ordered in non-increasing order, we deduce that the problem is maximized when $U = \mathrm{Id}_d$ and $V = \mathrm{Id}_{d'}$. This implies that $U_P = U_K$ and $V_P = V_K$, which concludes the proof. $\qquad\square$

Note that Lemma A4 is especially useful when the constraint of belonging to the set $\mathfrak{P}$ can be expressed as a constraint on the singular values. Observe that this is the case of $\mathbb{V}_{d'}(\mathbb{R}^d)$ since for all $P \in \mathbb{V}_{d'}(\mathbb{R}^d)$, we have $P^T P = \mathrm{Id}_{d'}$ and so an equivalent condition of belonging in $\mathbb{V}_{d'}(\mathbb{R}^d)$ is that $\boldsymbol{\sigma}(P) = \mathbb{1}_{d'}$.

### A.5 Some properties of symmetric matrices

Here we state two technical results on symmetric matrices that will be useful in the proofs of the results on Gaussian distributions.

**Lemma A5.** *Let $A \in \mathbb{S}^d$. We denote $\lambda_1$ and $\lambda_d$ its largest and smallest eigenvalues. For all $x \in \mathbb{R}^d$ such that $\|x\| = 1$, we have*

*(i) $x$ is an eigenvector of $A$ associated to $\lambda_1$ if and only if $x^T A x = \lambda_1$.*

*(ii) $x$ is an eigenvector of $A$ associated to $\lambda_d$ if and only if $x^T A x = \lambda_d$.*

*Proof.* Let $x \in \mathbb{R}^d$ such $\|x\| = 1$. Since $A$ is symmetric, there exists $O \in \mathbb{O}(\mathbb{R}^d)$ and $\Lambda = \mathrm{diag}((\lambda_k)_{1 \le k \le d})$ such that $x^T A x = x^T O \Lambda O^T x$. Denoting $z$ the vector $O^T x$, we get thus

$$x^T A x = z^T \Lambda z = \sum_{k=1}^d \lambda_k z_k^2 \ .$$

Hence it follows that

$$\lambda_d \|z\|^2 \le x^T A x \le \lambda_1 \|z\|^2 \ ,$$

with equality if and only if $z$ is an eigenvector associated with $\lambda_1$ or $\lambda_d$. $\qquad\square$

**Lemma A6.** *Suppose that $d \geq d'$. Let $\Sigma$ be a positive semi-definite (PSD) matrix of size $d + d'$ of the form*

$$\Sigma = \begin{pmatrix} \Sigma_0 & K \\ K^T & \Sigma_1 \end{pmatrix} ,$$

*with $\Sigma_0 \in \mathbb{S}^d_{++}$, $\Sigma_1 \in \mathbb{S}^{d'}_+$ and $K$ being a rectangular matrix of size $d \times d'$. Let $S = \Sigma_1 - K^T \Sigma_0^{-1} K$ be the Schur complement of $\Sigma$. Then there exists $r \leq d'$ and $B_r \in \mathbb{V}_r(\mathbb{R}^d)$ such that*

$$K = \Sigma_0^{\frac{1}{2}} B_r \Lambda_r U_r^T ,$$

*where $U_r \in \mathbb{V}_r(\mathbb{R}^{d'})$ and $\Lambda_r$ is a diagonal positive matrix of size $r$ such that*

$$\Sigma_1 - S = U_r \Lambda_r^2 U_r^T .$$

*Proof.* For a given Schur complement $S = \Sigma_1 - K^T \Sigma_0^{-1} K$, we have $K^T \Sigma_0^{-1} K = \Sigma_1 - S$. Since $\Sigma_0 \in \mathbb{S}^d_{++}$, we can deduce that $K^T \Sigma_0^{-1} K \in \mathbb{S}^{d'}_+$ and so that $\Sigma_1 - S \in \mathbb{S}^{d'}_+$. We note $r$ the rank of $K^T \Sigma_0^{-1} K$. One can observe that

$$r \leq d' \leq d ,$$

where the left-hand side inequality follows from the fact that $\mathrm{rk}(AB) \leq \min\{\mathrm{rk}(A), \mathrm{rk}(B)\}$. Then, $\Sigma_1 - S$ can be diagonalized

$$\Sigma_1 - S = K^T \Sigma_0^{-1} K = U\Lambda^2 U^T = U_r \Lambda_r^2 U_r^T , \tag{11}$$

with $\Lambda^2 = \mathrm{diag}(\lambda_1^2, ..., \lambda_r^2)^{[d']}$, $\Lambda_r^2 = \mathrm{diag}(\lambda_1^2, ..., \lambda_r^2)$, and $U_r \in \mathbb{V}_r(\mathbb{R}^{d'})$ such that $U = \begin{pmatrix} U_r & U_{d'-r} \end{pmatrix}$. From (11), we can deduce that

$$(\Sigma_0^{-\frac{1}{2}} K U_r \Lambda_r^{-1})^T \Sigma_0^{-\frac{1}{2}} K U_r \Lambda_r^{-1} = \mathrm{Id}_r ,$$

where $\Lambda_r$ is the unique PSD square-root of $\Lambda_r^2$. Let us set $B_r = \Sigma_0^{-\frac{1}{2}} K U_r \Lambda_r^{-1}$ such that $B_r \in \mathbb{V}_r(\mathbb{R}^d)$. It follows that

$$KU_r = \Sigma_0^{\frac{1}{2}} B_r \Lambda_r .$$

Moreover, since $U_{d-r}^T K^T \Sigma_0^{-1} K U_{d-r} = 0$ and $\Sigma_0 \in S_d^{++}(\mathbb{R})$, it follows that $KU_{d'-r} = 0$ and so

$$K = KUU^T = KU_r U_r^T = \Sigma_0^{\frac{1}{2}} B_r \Lambda_r U_r^T ,$$

which concludes the proof. $\square$

# B  Proofs of the theoretical results

## B.1  Proof of Proposition 2

*Proof of Proposition 2.* Takatsu (2010) has shown that the space of Gaussian distributions $\mathcal{N}(\mathbb{R}^d)$ is a complete metric space when endowed with $W_2$. Moreover, $\mathcal{N}(\mathbb{R}^d)$ is separable since it is a subspace of $\mathcal{W}_2(\mathbb{R}^d)$ which is itself a separable metric space when endowed with $W_2$ (Bolley, 2008). Thus, $\mathcal{N}(\mathbb{R}^d)$ is Polish and we can directly apply the Gromov-Wasserstein theory developed in Sturm (2012). Let $(\mathcal{N}(\mathbb{R}^d), W_2, \tilde{\mu})$ and $(\mathcal{N}(\mathbb{R}^{d'}), W_2, \tilde{\nu})$ be two metric measure spaces respectively in $\mathbb{M}_4$. Let us define

$$D(\tilde{\mu}, \tilde{\nu}) = \inf_{\pi \in \Pi(\tilde{\mu}, \tilde{\nu})} \int_{\mathcal{N}(\mathbb{R}^d) \times \mathcal{N}(\mathbb{R}^{d'})} \int_{\mathcal{N}(\mathbb{R}^d) \times \mathcal{N}(\mathbb{R}^{d'})} |W_2^2(\gamma, \gamma') - W_2^2(\zeta, \zeta')|^2 \mathrm{d}\pi(\gamma, \zeta) \mathrm{d}\pi(\gamma', \zeta') .$$

Applying Sturm (2012, Corollary 9.3), we get that $D$ defines a metric over the space of metric measure spaces of the form $(\mathcal{N}(\mathbb{R}^d), W_2, \tilde{\mu})$ quotiented by the strong isomorphisms, and thus we get directly that $D$ is symmetric, non-negative, satisfies the triangle inequality and $D(\tilde{\mu}, \tilde{\nu}) = 0$ if and only if there exists a bijection $\phi \colon \mathrm{supp}(\tilde{\mu}) \to \mathrm{supp}(\tilde{\nu})$ such that $\tilde{\nu} = \phi_{\#} \tilde{\mu}$, where for any $\gamma$ and $\gamma'$ in $\mathrm{supp}(\tilde{\mu})$, $W_2(\phi(\gamma), \phi(\gamma')) = W_2(\gamma, \gamma')$.

Now observe that if $\mu = \sum_k a_k \mu_k$ and $\nu = \sum_l b_l \nu_l$ are respectively in $GMM_K(\mathbb{R}^d)$ and $GMM_L(\mathbb{R}^{d'})$ and $\tilde{\mu} = \sum_k a_k \delta_{\mu_k}$ and $\tilde{\nu} = \sum_l b_l \delta_{\nu_l}$ are respectively in $\mathcal{P}(\mathcal{N}(\mathbb{R}^d))$ and $\mathcal{P}(\mathcal{N}(\mathbb{R}^{d'}))$, we have

$$\int_{\mathcal{N}(\mathbb{R}^d) \times \mathcal{N}(\mathbb{R}^d)} W_2^4(\gamma, \gamma') \mathrm{d}\tilde{\mu}(\gamma) \mathrm{d}\tilde{\mu}(\gamma') = \sum_{k,i} a_k a_i W_2^4(\mu_k, \mu_i) < +\infty ,$$

and

$$\int_{\mathcal{N}(\mathbb{R}^{d'}) \times \mathcal{N}(\mathbb{R}^{d'})} W_2^4(\zeta, \zeta') \mathrm{d}\tilde{\nu}(\zeta) \mathrm{d}\tilde{\nu}(\zeta') = \sum_{l,j} b_l b_j W_2^4(\nu_l, \nu_j) < +\infty ,$$

so $(\mathcal{N}(\mathbb{R}^d), W_2, \tilde{\mu})$ and $(\mathcal{N}(\mathbb{R}^{d'}), W_2, \tilde{\nu})$ are both in $\mathbb{M}_4$. Furthermore, we have $MGW_2(\mu, \nu) = D(\tilde{\mu}, \tilde{\nu})$. Hence $MGW_2$ inherits the metric properties of $D$, which concludes the proof. $\square$

## B.2 Proof of Proposition 3

*Proof of Proposition 3.* First recall that the push-foward measure $T_{\#}\mu$ with $\mu$ on $\mathbb{R}^{d'}$ and $T \colon \mathbb{R}^{d'} \to \mathbb{R}^d$ is defined as the measure on $\mathbb{R}^d$ such that for every Borel set $\mathsf{A}$ of $\mathbb{R}^d$, $T_{\#}\mu(\mathsf{A}) = \mu(T^{-1}(\mathsf{A}))$. Equivalently, for any measurable map $h \colon \mathbb{R}^d \to \mathbb{R}$, we have

$$\int_{\mathbb{R}^d} h(x) \mathrm{d}(T_{\#}\mu)(x) = \int_{\mathbb{R}^{d'}} (h \circ T)(y) \mathrm{d}\mu(y) .$$

Now observe that for any finite GMM $\mu$ on $\mathbb{R}^{d'}$ of the form $\mu = \sum_k^K a_k \mu_k$, we have

$$
\begin{aligned}
\int_{\mathbb{R}^{d'}} (h \circ T)(y) \mathrm{d}\mu(y) &= \int_{\mathbb{R}^{d'}} (h \circ T)(y) \mathrm{d}\left(\sum_k^K a_k \mu_k(y)\right) \\
&= \sum_k^K a_k \int_{\mathbb{R}^{d'}} (h \circ T)(y) \mathrm{d}\mu_k(y) \\
&= \sum_k^K a_k \int_{\mathbb{R}^d} h(x) \mathrm{d}(T_{\#}\mu_k)(x) \\
&= \int_{\mathbb{R}^d} h(x) \mathrm{d}\left(\sum_k^K a_k (T_{\#}\mu_k)(x)\right) ,
\end{aligned}
$$

and so $T_{\#}\mu$ is of the form $\sum_k^K a_k (T_{\#}\mu_k)$ with $T_{\#}\mu_k$ Gaussian since $T$ is necessarily affine as a consequence of Lemma A2. Thus, $T_{\#}\mu$ is in $GMM_\infty(\mathbb{R}^d)$. This proves that $\phi_T$ takes its values only in $GMM_\infty(\mathbb{R}^d)$ and that $\phi_T(\sum_{k=1}^K a_k \mu_k)$ is of the form $\sum_{k=1} a_k \nu_k$. Now observe that, for every $k$ and $i$ smaller than $K$,

$$W_2^2(\phi_T(\mu_k), \phi_T(\mu_i)) = \inf_{\pi \in \Pi(T_{\#}\mu_k, T_{\#}\mu_i)} \int_{\mathbb{R}^d \times \mathbb{R}^d} \|x - y\|^2 \mathrm{d}\pi(x, y) .$$

Using two times successively Lemma A1 using the fact that $T$ is an isometry an so for any $y \in \mathbb{R}^{d'}$, $\|T(y)\| = \|y\|$, it follows

$$\inf_{\pi \in \Pi(T_{\#}\mu_k, T_{\#}\mu_i)} \int_{\mathbb{R}^d \times \mathbb{R}^d} \|x - x'\|^2 \mathrm{d}\pi(x, x') = \inf_{\pi \in \Pi(\mu_k, \mu_i)} \int_{\mathbb{R}^{d'} \times \mathbb{R}^{d'}} \|y - y'\|^2 \mathrm{d}\pi(y, y') = W_2(\mu_k, \mu_i) .$$

Thus, $MGW_2(\mu, T_{\#}\mu) = 0$ as a direct consequence of Proposition 2, which concludes the proof. $\square$

## B.3 Proof of Proposition 7

We prove Proposition 7 before proving Proposition 6 because we will use the former in the proof of the latter.

*Proof of Proposition 7.* Since we suppose $d \geq d'$, we have

$$EW_2^2(\mu, \nu) = \inf_{\phi \in \mathrm{Isom}_{d'}(\mathbb{R}^d)} W_2^2(\mu, \phi_{\#}\nu) .$$

Let $\phi \in \mathrm{Isom}_{d'}(\mathbb{R}^d)$ for the Euclidean norm. Using Lemma A2, we get that there exists $P \in \mathbb{V}_{d'}(\mathbb{R}^d)$ and $b \in \mathbb{R}^d$ such that for all $y \in \mathbb{R}^{d'}$, $\phi(y) = Py + b$. Moreover, we have, using Lemma A1,

$$
\begin{aligned}
EW_2^2(\mu, \nu) &= \inf_{\phi \in \mathrm{Isom}_{d'}(\mathbb{R}^d)} \inf_{\pi \in \Pi(\mu, \phi_\# \nu)} \int_{\mathbb{R}^d \times \mathbb{R}^d} \|x - y\|^2 \mathrm{d}\pi(x, y) \\
&= \inf_{\phi \in \mathrm{Isom}_{d'}(\mathbb{R}^d)} \inf_{\pi \in \Pi(\mu, \nu)} \int_{\mathbb{R}^{d'} \times \mathbb{R}^d} \|x - \phi(y)\|^2 \mathrm{d}\pi(x, y) \\
&= \inf_{\pi \in \Pi(\mu, \nu)} \inf_{P \in \mathbb{V}_{d'}(\mathbb{R}^d), \ b \in \mathbb{R}^d} \int_{\mathbb{R}^{d'} \times \mathbb{R}^d} \|x - Py - b\|^2 \mathrm{d}\pi(x, y) \ ,
\end{aligned}
$$

which proves Equation (7). Now we show the equivalence with Problem (∗-COV). Using Lemma A3, Problem (7) can be rewritten

$$
\begin{aligned}
EW_2^2(\mu, \nu) &= \inf_{P \in \mathbb{V}_{d'}(\mathbb{R}^d)} \inf_{\pi \in \Pi(\bar{\mu}, \bar{\nu})} \int_{\mathbb{R}^d \times \mathbb{R}^{d'}} \|x - Py\|^2 \mathrm{d}\pi(x, y) \\
&= \inf_{P \in \mathbb{V}_{d'}(\mathbb{R}^d)} \inf_{\pi \in \Pi(\bar{\mu}, \bar{\nu})} \int_{\mathbb{R}^d \times \mathbb{R}^{d'}} \left( \|x\|^2 + \|Py\|^2 - 2\langle x, Py \rangle \right) \mathrm{d}\pi(x, y) \ .
\end{aligned}
$$

Since for all $P \in \mathbb{V}_{d'}(\mathbb{R}^d)$, $\|Py\|$ doesn't depend on $P$, we get that the problem is equivalent to

$$
\sup_{P \in \mathbb{V}_{d'}(\mathbb{R}^d)} \sup_{\pi \in \Pi(\bar{\mu}, \bar{\nu})} \int_{\mathbb{R}^d \times \mathbb{R}^{d'}} \langle x, Py \rangle \mathrm{d}\pi(x, y) \ .
$$

Now observe that for all $\pi \in \Pi(\bar{\mu}, \bar{\nu})$,

$$
\int_{\mathbb{R}^d \times \mathbb{R}^{d'}} \langle x, Py \rangle \mathrm{d}\pi(x, y) = \int_{\mathbb{R}^d \times \mathbb{R}^{d'}} \mathrm{tr}(xy^T P^T) \mathrm{d}\pi(x, y) = \int_{\mathbb{R}^d \times \mathbb{R}^{d'}} \mathrm{tr}(P^T xy^T) \mathrm{d}\pi(x, y) \ ,
$$

where we used the cyclical permutation property of the trace operator. Finally using the linearity of the trace, we get that the problem is equivalent to

$$
\sup_{P \in \mathbb{V}_{d'}(\mathbb{R}^d)} \sup_{\pi \in \Pi(\bar{\mu}, \bar{\nu})} \mathrm{tr}\left( P^T \int_{\mathbb{R}^d \times \mathbb{R}^{d'}} xy^T \mathrm{d}\pi(x, y) \right) \ ,
$$

or equivalently,

$$
\sup_{P \in \mathbb{V}_{d'}(\mathbb{R}^d)} \sup_{\pi \in \Pi(\bar{\mu}, \bar{\nu})} \left\langle P, \int_{\mathbb{R}^d \times \mathbb{R}^{d'}} xy^T \mathrm{d}\pi(x, y) \right\rangle \ .
$$

Now, using Lemma A4 and using the fact that if $P \in \mathbb{V}_{d'}(\mathbb{R}^d)$, $\boldsymbol{\sigma}(P) = \mathbb{1}_{d'}$, we get that the problem reduces to

$$
\sup_{\pi \in \Pi(\bar{\mu}, \bar{\nu})} \left\| \int_{\mathbb{R}^d \times \mathbb{R}^{d'}} xy^T \mathrm{d}\pi(x, y) \right\|_* \ ,
$$

and this is achieved for $P^* = U_\pi \, \mathrm{Id}_{d'}^{[d, d']} V_\pi^T$, where $U_\pi \in \mathbb{O}(\mathbb{R}^d)$ and $V_\pi \in \mathbb{O}(\mathbb{R}^{d'})$ are respectively the left and right orthogonal matrices of the SVD of $\int_{\mathbb{R}^d \times \mathbb{R}^{d'}} xy^T \mathrm{d}\pi(x, y)$, which concludes the proof. $\qquad \square$

## B.4 Proof of Proposition 6

Before turning to the proof of Proposition 6, we will prove two useful results. First, we show that the $EW_2$ problem is always achieved at an optimal couple $(\pi^*, \phi^*)$.

**Lemma B1.** *Let $\mu \in \mathcal{W}_2(\mathbb{R}^d)$ and $\nu \in \mathcal{W}_2(\mathbb{R}^{d'})$ and let suppose $d \geq d'$. Then there exists an optimal isometry $\phi^* \colon \mathbb{R}^{d'} \to \mathbb{R}^d$ such that $EW_2(\mu, \nu) = W_2(\mu, \phi_\#^* \nu)$.*

*Proof.* Using Lemma A3 and Lemma A1, we have that

$$
EW_2^2(\mu, \nu) = \inf_{P \in \mathbb{V}_{d'}(\mathbb{R}^d)} \inf_{\pi \in \Pi(\bar{\mu}, \bar{\nu})} \int_{\mathbb{R}^d \times \mathbb{R}^{d'}} \|x - Py\|^2 \mathrm{d}\pi(x, y)
$$

$$= \inf_{P \in \mathbb{V}_{d'}(\mathbb{R}^d)} W_2^2(\bar{\mu}, P_{\#}\bar{\nu}) \,,$$

where $\bar{\mu}$ and $\bar{\nu}$ are the centered measures associated with $\mu$ and $\nu$. Let us denote $J \colon P \mapsto W_2(\bar{\mu}, P_{\#}\bar{\nu})$ and let us show that $J$ is continuous. For any $P_0$ and $P_1$ in $\mathbb{V}_{d'}(\mathbb{R}^d)$, we have,

$$|J(P_0) - J(P_1)| = |W_2(\bar{\mu}, P_{0\#}\bar{\nu}) - W_2(\bar{\mu}, P_{1\#}\bar{\nu})| \leq W_2(P_{0\#}\bar{\nu}, P_{1\#}\bar{\nu}) \,,$$

where we used the triangular inequality property of $W_2$. Furthermore,

$$W_2^2(P_{0\#}\bar{\nu}, P_{1\#}\bar{\nu}) = \inf_{\pi \in \Pi(P_{0\#}\bar{\nu}, P_{1\#}\bar{\nu})} \int_{\mathbb{R}^d \times \mathbb{R}^d} \|x - y\|^2 \mathrm{d}\pi(x, y)$$

$$= \inf_{\pi \in \Pi(\bar{\nu}, \bar{\nu})} \int_{\mathbb{R}^{d'} \times \mathbb{R}^{d'}} \|P_0 x - P_1 y\|^2 \mathrm{d}\pi(x, y) \,,$$

where we used Lemma A1 twice. Now observe that the coupling $(\mathrm{Id}_{d'}, \mathrm{Id}_{d'})_{\#}\bar{\nu}$ is in $\Pi(\bar{\nu}, \bar{\nu})$, so it follows

$$\inf_{\pi \in \Pi(\bar{\nu}, \bar{\nu})} \int_{\mathbb{R}^{d'} \times \mathbb{R}^{d'}} \|P_0 x - P_1 y\|^2 \mathrm{d}\pi(x, y) \leq \int_{\mathbb{R}^{d'}} \|P_0 x - P_1 x\|^2 \mathrm{d}\bar{\nu}(x) \,.$$

Finally, for any $x \in \mathbb{R}^{d'}$, we have

$$\|P_0 x - P_1 x\|^2 \leq \|x\|^2 \sup_{\|z\|=1} \|(P_0 - P_1)z\|^2 \leq \|P_0 - P_1\|_{\mathcal{F}}^2 \|x\|^2 \,,$$

and so it follows that

$$|J(P_0) - J(P_1)|^2 \leq \|P_0 - P_1\|_{\mathcal{F}}^2 \int_{\mathbb{R}^n} \|x\|^2 \mathrm{d}\bar{\nu} \,.$$

Since $\nu$ is in $\mathcal{W}_2(\mathbb{R}^{d'})$, $\bar{\nu}$ is in $\mathcal{W}_2(\mathbb{R}^{d'})$ and so $\int_{\mathbb{R}^{d'}} \|x\|^2 \mathrm{d}\bar{\nu} < +\infty$. It follows that $|J(P_0) - J(P_1)| \longrightarrow 0$ when $\|P_0 - P_1\|_{\mathcal{F}}^2 \longrightarrow 0$ and so $J$ is continuous. Moreover, since $\mathbb{V}_{d'}(\mathbb{R}^d)$ is compact (James, 1976), $J$ has a minimum on $\mathbb{V}_{d'}(\mathbb{R}^d)$ as a result of the classic Weierstrass theorem that states that any real-valued continous function defined on a compact set achieves its infimum. Thus, there exists $P^*$ such that $EW_2(\mu, \nu) = W_2(\bar{\mu}, P_{\#}^*\bar{\nu})$ and setting $b^* = \mathbb{E}_{X \sim \mu}[X] - P^* \mathbb{E}_{Y \sim \nu}[Y]$ and $\phi^*(x) = P^* x + b^*$ for all $x \in \mathbb{R}^d$, we get that there exists $\phi^* \in \mathrm{Isom}_{d'}(\mathbb{R}^d)$ such that $EW_2(\mu, \nu) = W_2(\mu, \phi_{\#}^*\nu)$, which concludes the proof. $\qquad \square$

Now we show the following results, which imply that $EW_2$ remains unchanged when one of the two measures is immersed in a third Euclidean space of greater dimension than $d$ and $d'$.

**Lemma B2.** *Let $\mu \in \mathcal{W}_2(\mathbb{R}^d)$ and $\nu \in \mathcal{W}_2(\mathbb{R}^{d'})$ with $d$ not necessarily greater than $d'$. Let $r \geq \max\{d, d'\}$ and let $\psi \in \mathrm{Isom}_d(\mathbb{R}^r)$. Then, $EW_2(\mu, \nu) = EW_2(\psi_{\#}\mu, \nu)$.*

*Proof.* First, using Lemma A2, we get that there exists $P_1 \in \mathbb{V}_d(\mathbb{R}^r)$ and $b_1 \in \mathbb{R}^r$ such that for all $x \in \mathbb{R}^d$, $\psi(x) = P_1 x + b_1$. Since $r \geq d'$, we have, denoting $\bar{\mu}$, $\overline{\psi_{\#}\mu}$ and $\bar{\nu}$ the centered measures respectively associated with $\mu$, $\psi_{\#}\mu$, and $\nu$, and using successively Lemma A3 and Lemma A1,

$$EW_2^2(\psi_{\#}\mu, \nu) = \inf_{\pi \in \Pi(\psi_{\#}\mu, \nu)} \inf_{P \in \mathbb{V}_{d'}(\mathbb{R}^r),\ b \in \mathbb{R}^r} \int_{\mathbb{R}^r \times \mathbb{R}^{d'}} \|z - Py - b\|^2 \mathrm{d}\pi(z, y)$$

$$= \inf_{\pi \in \Pi(\overline{\psi_{\#}\mu}, \bar{\nu})} \inf_{P \in \mathbb{V}_{d'}(\mathbb{R}^r)} \int_{\mathbb{R}^r \times \mathbb{R}^{d'}} \|z - Py\|^2 \mathrm{d}\pi(z, y)$$

$$= \inf_{\pi \in \Pi(\bar{\mu}, \bar{\nu})} \inf_{P \in \mathbb{V}_{d'}(\mathbb{R}^r)} \int_{\mathbb{R}^d \times \mathbb{R}^{d'}} \|P_1 x - Py\|^2 \mathrm{d}\pi(x, y)$$

$$= \int_{\mathbb{R}^d} \|P_1 x\|^2 \mathrm{d}\bar{\mu}(x) + \int_{\mathbb{R}^{d'}} \|Py\|^2 \mathrm{d}\bar{\nu}(y) - 2 \sup_{\pi \in \Pi(\bar{\mu}, \bar{\nu})} \sup_{P \in \mathbb{V}_{d'}(\mathbb{R}^r)} \mathrm{tr}(P^T P_1 K_\pi)$$

$$= \int_{\mathbb{R}^d} \|x\|^2 \mathrm{d}\bar{\mu}(x) + \int_{\mathbb{R}^{d'}} \|y\|^2 \mathrm{d}\bar{\nu}(y) - 2 \sup_{\pi \in \Pi(\bar{\mu}, \bar{\nu})} \sup_{P \in \mathbb{V}_{d'}(\mathbb{R}^r)} \mathrm{tr}(P^T P_1 K_\pi) \,,$$

where $K_\pi = \int_{\mathbb{R}^d \times \mathbb{R}^{d'}} xy^T \mathrm{d}\pi(x, y)$. Using the equivalence with Problem ($*$-COV), we get

$$\sup_{\pi \in \Pi(\tilde{\mu}, \tilde{\nu})} \sup_{P \in \mathbb{V}_{d'}(\mathbb{R}^r)} \mathrm{tr}(P^T P_1 K_\pi) = \sup_{\pi \in \Pi(\tilde{\mu}, \tilde{\nu})} \|P_1 K_\pi\|_* .$$

Now observe that $P_1 K_\pi$ has the same singular values as $K_\pi$ since $K_\pi^T P_1^T P_1 K_\pi = K_\pi^T K_\pi$. Thus $\|P_1 K_\pi\|_* = \|K_\pi\|_*$ and so $EW_2(\psi_\# \mu, \nu) = EW_2(\mu, \nu)$, which concludes the proof. $\qquad\square$

Observe that Lemma B2 highlights that $EW_2$ shares close connections with the distance between metric measure spaces introduced in Sturm (2006) and defined in Equation (6). However it is not clear whether the two distances are strictly equivalent or not because the infimum in $\mathcal{Z}$ in Equation (6) also includes non-Euclidean spaces. However, if we restrict the problem only to Euclidean spaces $\mathcal{Z}$, then Lemma B2 directly implies that the two distances are equivalent. Now we are ready to prove Proposition 6.

*Proof of Proposition 6.* First observe that non-negativity is straightforward. Furthermore, observe also that if $d \neq d'$, symmetry is also straightfoward. Now suppose $d = d'$ and observe that that the set $\mathbb{V}_{d'}(\mathbb{R}^d)$ coincides with the set of orthogonal matrices $\mathbb{O}(\mathbb{R}^d)$. Thus we have

$$\begin{aligned}
\inf_{\phi \in \mathrm{Isom}_d(\mathbb{R}^d)} W_2(\mu, \phi_\# \nu) &= \inf_{\pi \in \Pi(\mu, \nu)} \inf_{P \in \mathbb{O}(\mathbb{R}^d), \ b \in \mathbb{R}^d} \int_{\mathbb{R}^d \times \mathbb{R}^d} \|x - Py - b\|^2 \mathrm{d}\pi(x, y) \\
&= \inf_{\pi \in \Pi(\mu, \nu)} \inf_{P \in \mathbb{O}(\mathbb{R}^d), \ b \in \mathbb{R}^d} \int_{\mathbb{R}^d \times \mathbb{R}^d} \|P^T x - y - P^T b\|^2 \mathrm{d}\pi(x, y) \\
&= \inf_{\psi \in \mathrm{Isom}_d(\mathbb{R}^d)} W_2(\psi_\# \mu, \nu) ,
\end{aligned}$$

and so $EW_2$ is also symmetric in that case. Before turning to the proof of the two other points, we recall that the infimum in $\phi$ is always achieved, see Lemma B1.

(i) Now we prove the triangle inequality. Let $r \geq \max\{d, d', d''\}$, $\phi_0 \in \mathrm{Isom}_d(\mathbb{R}^r)$ and for $\xi \in \mathcal{W}_2(\mathbb{R}^{d''})$, let $\phi_1 \in \arg\min_{\phi \in \mathrm{Isom}_{d''}(\mathbb{R}^r)} W_2(\phi_{0\#} \mu, \phi_\# \xi)$. We have, using first Lemma B2, then using the triangle inequality property of $W_2$,

$$\begin{aligned}
EW_2(\mu, \nu) = EW_2(\phi_{0\#} \mu, \nu) &= \inf_{\phi \in \mathrm{Isom}_{d'}(\mathbb{R}^r)} W_2(\phi_{0\#} \mu, \phi_\# \nu) \\
&\leq \inf_{\phi \in \mathrm{Isom}_{d'}(\mathbb{R}^r)} \left[ W_2(\phi_{0\#} \mu, \phi_{1\#} \xi) + W_2(\phi_{1\#} \xi, \phi_\# \nu) \right] \\
&\leq W_2(\phi_{0\#} \mu, \phi_{1\#} \xi) + \inf_{\phi \in \mathrm{Isom}_{d'}(\mathbb{R}^r)} W_2(\phi_{1\#} \xi, \phi_\# \nu) \\
&\leq EW_2(\phi_{0\#} \mu, \xi) + EW_2(\phi_{1\#} \xi, \nu) .
\end{aligned}$$

We conclude then by applying Lemma B2 on both terms.

(ii) Suppose without any loss of generality that $d \geq d'$ and suppose $EW_2(\mu, \nu) = 0$. Since the infimum in $\phi$ is achieved, there exists $\phi \in \mathrm{Isom}_{d'}(\mathbb{R}^d)$ such that $W_2(\mu, \phi_\# \nu) = 0$ and so $\mu = \phi_\# \nu$. The reverse implication is obvious.

Finally, observe that if $\mu$ and $\nu$ have finite order 2 moments, then $EW_2$ necessarily takes finite values, and so $EW_2$ defines a pseudometric on $\bigsqcup_{k \geq 1} \mathcal{W}_2(\mathbb{R}^k)$. $\qquad\square$

## B.5 Proof of Proposition 8

*Proof of Proposition 8.* As seen above, Problem ($EW_2$) is equivalent to

$$\sup_{\pi \in \Pi(\mu, \nu)} \sup_{P \in \mathbb{V}_{d'}(\mathbb{R}^d)} \langle P, K_\pi \rangle_\mathcal{F} ,$$

where $K_\pi = \int xy^T \mathrm{d}\pi(x,y)$. As in Delon et al. (2022a), we use the necessary condition for $\pi$ to be in $\Pi(\mu,\nu)$ that is that the covariance matrix $\Sigma_\pi$ of the law $\pi$ is a PSD matrix, or equivalently that the Schur complement of $\Sigma_\pi$, i.e. $\Sigma_1 - K_\pi^T \Sigma_0^{-1} K_\pi$ is also a PSD matrix. This gives the following inequality:

$$\sup_{\pi \in \Pi(\mu,\nu)} \sup_{P \in \mathbb{V}_{d'}(\mathbb{R}^d)} \langle P, K_\pi \rangle_\mathcal{F} \leq \max_{K \,:\, \Sigma_1 - K^T \Sigma_0^{-1} K \in \mathbb{S}_+^{d'}} \max_{P \in \mathbb{V}_{d'}(\mathbb{R}^d)} \langle P, K \rangle_\mathcal{F} \,.$$

The rest of the proof is inspired from the proof of the closed-form of the $W_2$ between two Gaussians provided by Givens et al. (1984). We want to solve the following constrained optimization problem

$$\min_{\substack{\Sigma_1 - K^T \Sigma_0^{-1} K \in \mathbb{S}_+^{d'} \\ P \in \mathbb{V}_{d'}(\mathbb{R}^d)}} -2\mathrm{tr}(P^T K) \,.$$

Using Lemma A6, we can write $\mathrm{tr}(P^T K)$ as a function of $B_r$. This gives the following equivalent constrained optimization problem

$$\min_{B_r^T B_r = \mathrm{Id}_r, P^T P = \mathrm{Id}_{d'}} -2\mathrm{tr}(P^T \Sigma_0^{\frac{1}{2}} B_r \Lambda_r U_r^T) \,.$$

The Lagrangian of this latter problem reads as

$$\mathcal{L}(B_r, P, C_0, C_1) = -2\mathrm{tr}(P^T \Sigma_0^{\frac{1}{2}} B_r \Lambda_r U_r^T) + \mathrm{tr}(C_0(B_r^T B_r - \mathrm{Id}_r)) + \mathrm{tr}(C_1(P^T P - \mathrm{Id}_{d'})) \,,$$

where $C_0 \in \mathbb{S}^r$ and $C_1 \in \mathbb{S}^{d'}$ are the Lagrange multipliers respectively associated with the constraints $B_r^T B_r = \mathrm{Id}_r$ and $P^T P = \mathrm{Id}_{d'}$. The first order condition gives

$$\begin{cases} \Sigma_0^{\frac{1}{2}} P U_r \Lambda_r = B_r C_0 \\ \Sigma_0^{\frac{1}{2}} B_r \Lambda_r U_r^T = P C_1 \,. \end{cases}$$

Since $\Sigma_0$, $P$, $U_r$, and $\Lambda_r$ are full rank, $\Sigma_0^{\frac{1}{2}} P U_r \Lambda_r$ is of rank $r$ and so $C_0$ is also of rank $r$. Thus we get that

$$B_r = \Sigma_0^{\frac{1}{2}} P U_r \Lambda_r C_0^{-1} \,,$$

and so

$$B_r^T B_r = \mathrm{Id}_r = C_0^{-1} \Lambda_r U_r^T P^T \Sigma_0 P U_r \Lambda_r C_0^{-1} \,.$$

Thus,

$$C_0 = (\Lambda_r U_r^T P^T \Sigma_0 P U_r \Lambda_r)^{\frac{1}{2}} \,.$$

On the other hand, by reinjecting the expression of $B_r$ in the other first order condition we get

$$P^T \Sigma_0 P U_r \Lambda_r (\Lambda_r U_r^T P^T \Sigma_0 P U_r \Lambda_r)^{-\frac{1}{2}} \Lambda_r U_r^T = C_1 \,.$$

By multiplying this equation by itself we get

$$P^T \Sigma_0 P U_r \Lambda_r^2 U_r^T = C_1^2 \,.$$

Since $C_1^2$ is symmetric we get that $P^T \Sigma_0 P$ commutes with $U_r \Lambda_r^2 U_r^T$ and so $\Sigma_1 - S$. Moreover, as before we have

$$\mathrm{tr}(P^T K) = \mathrm{tr}(((\Sigma_1 - S)^{\frac{1}{2}} P^T \Sigma_0 P (\Sigma_1 - S)^{\frac{1}{2}})^{\frac{1}{2}})$$
$$= \mathrm{tr}((\Sigma_1 - S)^{\frac{1}{2}} (P^T \Sigma_0 P)^{\frac{1}{2}}) \,.$$

Using the Courant-Fischer min-max theorem (Courant, 1920; Fischer, 1905) to characterize the eigenvalues of $\Sigma_1 - S$, see (Givens et al., 1984, Proposition 7) for details, we get that $\mathrm{tr}(P^T K)$ is maximized when $S = 0$ and so the problem is equivalent to the following problem

$$\max_{\substack{P \in \mathbb{V}_{d'}(\mathbb{R}^d) \\ P^T \Sigma_0 P \Sigma_1 = \Sigma_1 P^T \Sigma_0 P}} \mathrm{tr}(\hat{D}_1^{\frac{1}{2}} D_{0,P}^{\frac{1}{2}}) \,,$$

where $(\hat{P}_1, \hat{D}_1)$ is any diagonalization of $\Sigma_1$ and $D_{0,P} = \hat{P}_1^T P^T \Sigma_0 P \hat{P}_1$. For all $y \in \mathbb{R}^{d'}$ we have

$$\alpha_d \|y\|^2 \leq y^T P^T \Sigma_0 P y \leq \alpha_1 \|y\|^2 \ ,$$

where $\alpha_1, \ldots, \alpha_d$ are the eigenvalues of $\Sigma_0$ ordered in non-increasing order. Thus, denoting $\lambda_1, \ldots, \lambda_{d'}$ the eigenvalues of $P^T \Sigma_0 P$, we get that for all $k \leq d'$,

$$\alpha_d \leq \lambda_k \leq \alpha_1 \ .$$

Since we want to maximize $\text{tr}(\hat{D}_1^{\frac{1}{2}} D_{0,P}^{\frac{1}{2}})$, we set the largest eigenvalue $\lambda_1$ of $P^T \Sigma_0 P$ to $\alpha_1$. We denote $y_1 \in \mathbb{R}^{d'}$ the eigenvector associated. We have $y_1 P^T \Sigma_0 P y_1 = \alpha_1$ and $\|P y_1\| = \|y_1\| = 1$ so using Lemma A5, we get that $\|P y_1\|$ is an eigenvector of $\Sigma_0$ associated with $\alpha_1$. Let $\lambda_k$ and $y_k$ be any other eigenvalue and its associated eigenvector in the orthonormal basis in which $P^T \Sigma_0 P$ is diagonal. We have $y_k^T y_1 = 0$ and so $y_k^T P^T P y_1 = 0$. Thus $P y_k$ is orthogonal to $P y_1$. Since $\|P y_k\| = 1$, we get that $P y_k$ is also an eigenvector of $\Sigma_0$ and so it exists $i \leq d-1$ such that $\lambda_k = y_k^T P^T \Sigma_0 P y_k = \alpha_i$. Thus, we conclude that the eigenvalues of the optimal $P^T \Sigma_0 P$ are the $d'$ largest eigenvalues of $\Sigma_0$. Moreover, $\text{tr}(\hat{D}_1^{\frac{1}{2}} D_{0,P}^{\frac{1}{2}})$ is clearly maximized when $D_{0,P}$ and $\hat{D}_1$ have their eigenvalues sorted in the same order. We conclude then that setting $D_{0,P} = D_0^{(d')}$ and $\hat{D}_1 = D_1$, where $D_0$ and $D_1$ are the diagonal matrices associated with the diagonalizations that sort the eigenvalues in non-increasing, maximizes the problem and so it follows that

$$\max_{\substack{\Sigma_1 - K^T \Sigma_0^{-1} K \in \mathbb{S}_+^{d'} \\ P \in \mathbb{V}_{d'}(\mathbb{R}^d)}} 2\text{tr}(P^T K) = 2\text{tr}(D_0^{(d')\frac{1}{2}} D_1^{\frac{1}{2}}) \ .$$

Finally, observe that when setting $K^*$ of the form

$$K^* = P_0 (\widetilde{I}_{d'} D_0^{(d')\frac{1}{2}} D_1^{\frac{1}{2}})^{[d,d']} P_1^T \ ,$$

we have

$$\|K\|_* = \text{tr}((K^{*T} K^*)^{\frac{1}{2}}) = \text{tr}((D_0^{(d')} D_1)^{\frac{1}{2}}) = \text{tr}(D_0^{(d')\frac{1}{2}} D_1^{\frac{1}{2}}) \ .$$

Moreover, observe that this is the solution of Equation ($\mathcal{F}$-COV) exhibited in (Delon et al., 2022a, Lemma 3.2). Thus $K^*$ is cleary in the feasible set and so is optimal. By reinjecting the optimal value in the expression of $EW_2(\mu, \nu)$, we get

$$EW_2^2(\mu, \nu) = \text{tr}(D_0) + \text{tr}(D_1) - 2\text{tr}(D_0^{(d')\frac{1}{2}} D_1^{\frac{1}{2}}) \ .$$

Furthermore, using the results of Delon et al. (2022a), we get directly that the optimal plans $\pi^*$ are of the form $(\text{Id}_d, T)_{\#} \mu$ with $T$ linear of the form

$$T = P_1 \left( \widetilde{I}_{d'} D_1^{\frac{1}{2}} D_0^{d'-\frac{1}{2}} \right)^{[d',d]} P_0^T$$

Finally, observe that $K^*$ admits as SVD $P_0 (D_0^{(d')\frac{1}{2}} D_1^{\frac{1}{2}})^{[d,d']} \widetilde{I}_{d'} P_1^T$. For a given fixed $\widetilde{I}_{d'}$, we get using Lemma A4, that the optimal $P^*$ associated with $K^*$ is $P^* = P_0 \widetilde{I}_{d'}^{[d,d']} P_1^T$, which concludes the proof. □

## B.6 Proof of Lemma 10

*Proof of Lemma 10.* First note that in this proof, we denote $\mathbb{R}^{d \times d'}$ the set of matrices of size $d \times d'$ that we distinguish from the set $\mathbb{R}^{dd'}$ of vector with $d \times d'$ coordinates. We set $g \colon P \in \mathbb{R}^{d \times d'} \mapsto \Sigma_1^{\frac{1}{2}} P^T \Sigma_0 P \Sigma_1^{\frac{1}{2}}$ and $h \colon Q \in \mathbb{S}_+^{d'} \mapsto Q^{\frac{1}{2}}$ such that for all matrix $P$ of size $d \times d'$, we have

$$f(P) = \text{tr}(h(g(P))) \ .$$

For any matrix $A \in \mathbb{R}^{d \times d'}$, we denote $\mathrm{vec}(A) \in \mathbb{R}^{dd'}$ the vector obtained by stacking the columns of $A$. Observe that, see (Magnus and Neudecker, 2019) for details, for any function $\phi \colon \mathbb{R}^{d \times d'} \to \mathbb{R}^{r \times s}$, the Jacobian matrix $J[\phi]$ of $\phi$ can be defined as, for all $P \in \mathbb{R}^{d \times d'}$,

$$J[\phi](P) = \frac{\partial \mathrm{vec}(f(P))}{\partial \mathrm{vec}(P)} \; .$$

Moreover, observe that since $f \colon \mathbb{R}^{d \times d'} \to \mathbb{R}$, $J[f][P] \in \mathbb{R}^{dd'}$ and

$$\frac{\partial f(P)}{\partial P} = \mathrm{vec}^{-1}(J^T[f](P)) \; ,$$

where $\mathrm{vec}^{-1}$ is the inverse vector operator, i.e. such that for any $A \in \mathbb{R}^{d \times d'}$, $\mathrm{vec}^{-1}(\mathrm{vec}(A)) = A$ . Applying the chain rule to derive $f$, we have

$$J[f](P) = J[\mathrm{tr}]((h \circ g)(P)) J[h](g(P)) J[g](P) \; .$$

- First, we compute $J[g](P)$. It follows, using formula provided by Petersen et al. (2008) and Magnus and Neudecker (2019),

$$\partial(\Sigma_1^{\frac{1}{2}} P^T \Sigma_0 P \Sigma_1^{\frac{1}{2}}) = \Sigma_1^{\frac{1}{2}} \partial P^T \Sigma_0 P \Sigma_1^{\frac{1}{2}} + \Sigma_1^{\frac{1}{2}} P^T \Sigma_0 \partial P \Sigma_1^{\frac{1}{2}},$$

and so

$$
\begin{aligned}
\partial \mathrm{vec}(\Sigma_1^{\frac{1}{2}} P^T \Sigma_0 P \Sigma_1^{\frac{1}{2}}) &= (\Sigma_1^{\frac{1}{2}} P^T \Sigma_0 \otimes_K \Sigma_1^{\frac{1}{2}}) \partial \mathrm{vec}(P^T) + (\Sigma_1^{\frac{1}{2}} \otimes_K \Sigma_1^{\frac{1}{2}} P^T \Sigma_0) \partial \mathrm{vec}(P) \\
&= (\Sigma_1^{\frac{1}{2}} P^T \Sigma_0 \otimes_K \Sigma_1^{\frac{1}{2}}) K_{dd'} \partial \mathrm{vec}(P) + (\Sigma_1^{\frac{1}{2}} \otimes_K \Sigma_1^{\frac{1}{2}} P^T \Sigma_0) \partial \mathrm{vec}(P) \\
&= (I_{d'^2} + K_{d'^2})(\Sigma_1^{\frac{1}{2}} \otimes_K \Sigma_1^{\frac{1}{2}} P^T \Sigma_0) \partial \mathrm{vec}(P) \; ,
\end{aligned}
$$

where $\otimes_K$ denotes the Kronecker product and for any $r$, $K_r$ is the commutation matrix of size $r \times r$, see (Magnus and Neudecker, 2019) for details. Thus,

$$J[g](P) = (I_{d'^2} + K_{d'^2})(\Sigma_1^{\frac{1}{2}} \otimes_K \Sigma_1^{\frac{1}{2}} P^T \Sigma_0) \; .$$

- Now we compute $J[h](Q)$. Observe that we have for any $Q \in \mathbb{S}_+^{d'}$,

$$Q^{\frac{1}{2}} Q^{\frac{1}{2}} = Q \; .$$

Thus it follows, denoting $s \colon Q \mapsto Q^{\frac{1}{2}}$,

$$\partial s(Q) Q^{\frac{1}{2}} + Q^{\frac{1}{2}} \partial s(Q) = \partial Q \; .$$

This latter equation is a Sylvester equation with variable $\partial s(Q)$, which is equivalent to the following linear system:

$$(Q^{\frac{1}{2}} \oplus_K Q^{T \frac{1}{2}}) \partial \mathrm{vec}(s(Q)) = \partial \mathrm{vec}(Q) \; ,$$

where $\oplus_K$ stands for the Kronecker sum. If $Q$ is non-degenerate, $Q^{\frac{1}{2}} \oplus_K Q^{T \frac{1}{2}}$ is also non-degenerate and so in that case

$$J[h](Q) = (Q^{\frac{1}{2}} \oplus_K Q^{T \frac{1}{2}})^{-1} \; .$$

- Finally, it is easy to see that for $R \in \mathbb{R}^{d' \times d'}$ we have

$$J[\mathrm{tr}](R) = \mathrm{vec}^T(\mathrm{Id}_{d'}).$$

Thus, denoting $A = \Sigma_1^{\frac{1}{2}} P^T \Sigma_0 P \Sigma_1^{\frac{1}{2}}$ and observing that $A$ is symmetric and full-rank when $P$ is full-rank (since we supposed that $\Sigma_0$ and $\Sigma_1$ are full rank), it follows that for all full-rank matrix $P$ of size $d \times d'$,

$$J^T[f](P) = (\Sigma_1^{\frac{1}{2}} \otimes_K \Sigma_0 P \Sigma_1^{\frac{1}{2}})(I_{d'^2} + K_{d'^2})(A^{\frac{1}{2}} \oplus_K A^{\frac{1}{2}})^{-1}\mathrm{vec}(\mathrm{Id}_{d'}) \ ,$$

where we used that $K_{d'^2}$ and $(A \oplus_K A)^{-1}$ were symmetric. Observe now that $(A^{\frac{1}{2}} \oplus_K A^{\frac{1}{2}})^{-1}\mathrm{vec}(\mathrm{Id}_{d'}) = \mathrm{vec}(X)$, where $X \in \mathbb{R}^{d' \times d'}$ is the unique solution of the following Sylvester equation

$$A^{\frac{1}{2}} X + X A^{\frac{1}{2}} = \mathrm{Id}_{d'} \ .$$

Since $A$ is symmetric, one can set $A = QDQ^T$ where $Q \in \mathbb{O}(\mathbb{R}^{d'})$ and $D$ is a diagonal matrix of size $d'$. The Sylvester equation can be rewritten

$$D^{\frac{1}{2}} Y + Y D^{\frac{1}{2}} = \mathrm{Id}_{d'} \ ,$$

where $Y = Q^T X Q$. Since $A$ is full-rank, $D$ is invertible and it is easy to see that the unique solution of this latter equation is $Y = (1/2)D^{-\frac{1}{2}}$ and so $X = (1/2)A^{-\frac{1}{2}}$ and thus

$$(A^{\frac{1}{2}} \oplus_K A^{\frac{1}{2}})^{-1}\mathrm{vec}(\mathrm{Id}_{d'}) = \frac{1}{2}\mathrm{vec}(A^{-\frac{1}{2}}) \ .$$

Moreover, since $A$ is symmetric, we have $K_{d'^2}\mathrm{vec}(A^{-\frac{1}{2}}) = \mathrm{vec}(A^{-\frac{1}{2}})$ and so it follows that

$$J^T[f](P) = (\Sigma_1^{\frac{1}{2}} \otimes_K \Sigma_0 P \Sigma_1^{\frac{1}{2}})\mathrm{vec}(A^{-\frac{1}{2}})$$
$$= \mathrm{vec}(\Sigma_0 P \Sigma_1^{\frac{1}{2}} A^{-\frac{1}{2}} \Sigma_1^{\frac{1}{2}}) \ ,$$

which concludes the proof. $\qquad\square$

## C   More details on Projection Wasserstein discrepancy

In this section, we give more details on the difference between $EW_2$ and the OT distance introduced in Cai and Lim (2022) that we call here *projection Wasserstein discrepancy*. We recall that for $\mu \in \mathcal{W}_2(\mathbb{R}^d)$ and $\nu \in \mathcal{W}_2(\mathbb{R}^{d'})$ with $d \geq d'$, this OT distance is defined as

$$PW_2(\mu, \nu) = \inf_{\phi \in \Gamma_d(\mathbb{R}^{d'})} W_2(\phi_\# \mu, \nu) \ , \tag{$PW_2$}$$

where $\Gamma_d(\mathbb{R}^{d'})$ is the set of all affine mapping from $\mathbb{R}^d$ to $\mathbb{R}^{d'}$ of the form $\varphi(x) = P^T(x - b)$ with $P \in \mathbb{V}_{d'}(\mathbb{R}^d)$ and $b \in \mathbb{R}^d$. One key results of Cai and Lim (2022) is to show that $PW_2$ has the following equivalent formulation

$$PW_2(\mu, \nu) = \inf_{\xi \in \mathcal{W}_2^\nu(\mathbb{R}^d)} W_2(\mu, \xi) \ ,$$

where $\mathcal{W}_2^\nu(\mathbb{R}^d)$ is the subset of $\mathcal{W}_2(\mathbb{R}^d)$ defined as

$$\mathcal{W}_2^\nu(\mathbb{R}^d) = \{\xi \in \mathcal{W}_2(\mathbb{R}^d) : \ \text{there exists } \phi(x) = P^T(x - b) \text{ with } P \in \mathbb{V}_{d'}(\mathbb{R}^d) \text{ and } b \in \mathbb{R}^{d'} \text{ such that } \phi_\# \xi = \nu\} \ .$$

Observe that this latter formulation is structurally different of $EW_2$ since for any isometry $\phi \colon \mathbb{R}^{d'} \to \mathbb{R}^d$, the distribution $\phi_\# \nu$ is necessarily degenerate, whereas this is not the case for the distribution $\xi$. The difference between $EW_2$ and $PW_2$ is illustrated in Figure C1.

To highlight even more the difference between $EW_2$ and $PW_2$, we derive an equivalent problem of Problem ($PW_2$). Observe that in that case, the mapping $\phi$ in ($PW_2$) is not an isometry since it is not injective. As a result, the term that previously depended only on the marginal $\mu$ in the developpement of the square of the Euclidean distance will now depend on $P$. More precisely, this gives the following result.

**Proposition C1.** *Let $\mu \in \mathcal{W}_2(\mathbb{R}^d)$ and $\nu \in \mathcal{W}_2(\mathbb{R}^{d'})$ and let suppose $d \geq d'$. Problem ($PW_2$) is equivalent to*

$$\inf_{\pi \in \Pi(\bar{\mu}, \bar{\nu})} \inf_{P \in \mathbb{V}_{d'}(\mathbb{R}^d)} \left( \mathrm{tr}(P^T \Sigma_x P) - 2\mathrm{tr}(P^T K_\pi) \right) \ , \tag{12}$$

*where $\Sigma_x = \int_{\mathbb{R}^d \times \mathbb{R}^d} xx^T \mathrm{d}\bar{\mu}(x)$, $K_\pi = \int_{\mathbb{R}^d \times \mathbb{R}^{d'}} xy^T \mathrm{d}\pi(x, y)$, and where $\bar{\mu}$ and $\bar{\nu}$ are the centered measures associated with $\mu$ and $\nu$.*

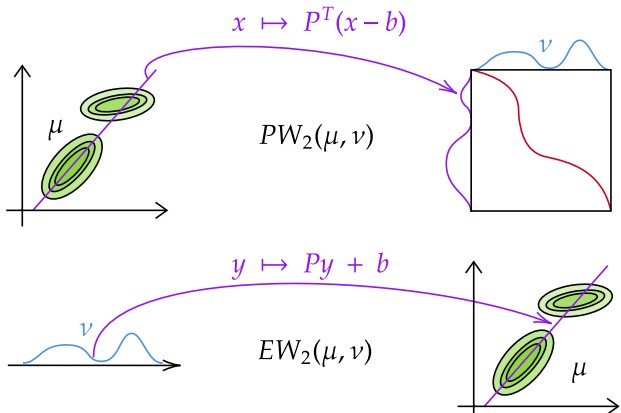

Figure C1: Link between $PW_2$ and $EW_2$ for two distributions $\mu$ and $\nu$ respectively on $\mathbb{R}^2$ and $\mathbb{R}$. In $PW_2$, $\mu$ is projected into $\mathbb{R}$ by a mapping of the form $x \mapsto P^T(x - b)$. In $EW_2$, $\nu$ is transformed into a degenerate measure (lying on the purple line) on $\mathbb{R}^2$ with an isometric mapping of the form $y \mapsto Py + b$.

*Proof of Proposition C1.* First observe that using Lemma A3, we can consider without any loss generality that $\mu$ and $\nu$ are centered and omit $b$. Using Lemma A1, it follows

$$PW_2^2(\mu, \nu) = \inf_{P \in \mathbb{V}_{d'}(\mathbb{R}^d)} \inf_{\pi' \in \Pi(P_\#^T \mu, \nu)} \int_{\mathbb{R}^{d'} \times \mathbb{R}^{d'}} \|z - y\|^2 \mathrm{d}\pi'(z, y)$$

$$= \inf_{P \in \mathbb{V}_{d'}(\mathbb{R}^d)} \inf_{\pi \in \Pi(\mu, \nu)} \int_{\mathbb{R}^d \times \mathbb{R}^{d'}} \|P^T x - y\|^2 \mathrm{d}\pi(x, y)$$

$$= \inf_{P \in \mathbb{V}_{d'}(\mathbb{R}^d)} \left( \int_{\mathbb{R}^d} \|P^T x\|^2 \mathrm{d}\mu(x) + \int_{\mathbb{R}^{d'}} \|y\|^2 \mathrm{d}\nu(y) - 2 \sup_{\pi \in \Pi(\mu, \nu)} \int_{\mathbb{R}^d \times \mathbb{R}^{d'}} (P^T x)^T y \mathrm{d}\pi(x, y) \right) ,$$

and so the problem is equivalent to

$$\inf_{P \in \mathbb{V}_{d'}(\mathbb{R}^d)} \left( \int_{\mathbb{R}^d} \|P^T x\|^2 \mathrm{d}\mu(x) - 2 \sup_{\pi \in \Pi(\mu, \nu)} \int_{\mathbb{R}^{d'} \times \mathbb{R}^{d'}} (P^T x)^T y \mathrm{d}\pi(x, y) \right) ,$$

which is itself equivalent to (12), which concludes the proof. $\qquad\square$

Observe that Problem (12) can be interpreted as a regularization in $P$ of the $EW_2$ problem since we have seen above that this latter was equivalent to the following problem

$$\sup_{\pi \in \Pi(\bar{\mu}, \bar{\nu})} \sup_{P \in \mathbb{V}_{d'}(\mathbb{R}^d)} \mathrm{tr}(P^T K_\pi) .$$

It can also be interpreted as a $W_2$ problem between $\nu$ and a measure $\mu'$ which has a different second-order moment than $\mu$.

