# OpenReview forum: "Gromov-Wasserstein-like Distances in the Gaussian Mixture Models Space"
_TMLR — Accepted by TMLR_

### Review · Reviewer_QjXg · 2024-05-24

**Summary Of Contributions:**

This paper investigates the optimal transport problem and first proposes a novel distance function to extend MW2. This distance, named $MGW_2$, is efficient for estimating distances between GMMs. Furthermore, the authors develop another distance function, the Embedded Wasserstein distance ($MEW_2$), which is also adapted to derive a distance as well as optimal transportation plans between GMMs. These OT distances are also by design particularly suited to settings where there already exists a kind of clustering structure in the data Finally, this paper conducts experiments to validate the efficiency of their methods. The results demonstrate the effectiveness of their method.
Although $MEW_2$ has some shortcomings in practical applications, $MGW_2$ remains effective.

**Audience:**

Yes

**Claims And Evidence:**

Yes

**Requested Changes:**

I advise the authors to divide the section 'Discussion' into conclusion and future work.

**Strengths And Weaknesses:**

Strengths:

1) The authors provide detailed proofs and the paper is well-written and easy to follow

2) These new distances exhibit certain innovation and practicality in handling Gaussian Mixture Models (GMMs) and demonstrate high efficiency under large-scale problems.

3) The experiments of this paper are well-developed and comprehensive, fully demonstrating the superiority of these two methods.



Weakness:

1) The innovation of this paper may be lacking. Can you highlight the difficulty in your proof?

2) Although the algorithms in this paper are primarily designed for large-scale problems, I am still uncertain about their performance on smaller-scale problems. Since the paper focuses on large-scale settings, even if the performance on smaller scales is not good, it does not affect my overall opinion of the paper.

---

### Review · Reviewer_BZ1F · 2024-06-03

**Summary Of Contributions:**

Computing the classical Wasserstein distance between probability distributions has two practical limitations: it is computationally expensive and prevents users from comparing distributions supported on two different probability spaces. This paper presents a way to bypass these drawbacks in the setting where the two probability distributions are (nearly) Gaussian Mixture Models (GMM). The paper has the following contributions:

- (Definition MGW2 distance) The authors combine ideas due to Chen et al. (2018) and Delon and Desolneux (2020) on the Wasserstein distance between GMM and to Memoli (2011) on the Gromov-Wasserstein distance to define a Gromov-Wasserstein distance for GMM, dubed MWG2. The main benefit of this distance is that it reduces to a simpler-looking (yet still hard-to-solve) problem (see Definition 1).
- The authors also investigate some of the basic properties of MGW2 (see section 3.1), but most of these follow directly from existing results.
- (Definition MEW2) The definition of MGW2 does not provide a direct way to extract a transportation map between distributions. So, the authors propose a second distance MEW2 (see Definition 8), which bypasses this issue.
- MGW2 and MEW2 yield non-convex problems, so the authors propose two-stage iterative methods to compute each one of these distances. In the first stage, they initialize using an annealing procedure inspired by a method by Alvarez-Melis et al. (2019) and then refine the solution using alternating minimization.

**Audience:**

Yes

**Claims And Evidence:**

Yes

**Requested Changes:**

- One of the critical goals of this work seems to be having an invariant distance (to certain transformations) between probability measures; how the abstract and the introduction are written makes it sound like a nice byproduct. Consider rewriting to emphasize the fact that you aimed for invariance.
- In the abstract, introduce the abbreviation GMMs (currently, it is used but not introduced).
- The title has a grammatical mistake it should be Wasserstein (note the s before the t) instead of Wassertein.
- P2 First line: footnotes should go after punctuation.
- P2 Change: "is invariant to important families of transformations" to "is invariant to a given family of transformations''
- P3 Notations: I would call this section Notation (without the s); the same goes for the first line of this section.
- P3 and P4 This is mostly personal preference, but consider changing the format of the notation section to a couple of paragraphs inline instead of a list. The list occupies too much space and does not match the style of the rest of the paper.
- P5 Add a reference for the fact that $MW_2$ defines a metric on $GGM_\infty(R^d)$.
- P8 Before Proposition 2: 'developped' -> developed.
- The P8 Question before Proposition 3 is hard to parse; consider rewriting.
- P8 The Euclidian norm is only one; it is defined in any dimension, but only one. Change the statement to say it is an isometry for the Euclidian norm.
- P8 Consider adding an Example environment for the example at the end of this page.
- P9 is hard to follow for the uninitiated; consider rewriting. Explicitly introduce $\pi*$ and $\omega*$. Don't call a matrix $Id$ if it is not the identity. Consider dropping the notation $A^{[d', d]}$ and simply write it explicitly in the definition of $T_{GGW_2}^{k, l}$.
- All of Page 9 feels unnecessary; it could be compressed to a paragraph at best. The authors are explaining how not to do things, which feels distracting.
- Page 11 Section 4: "explicit'' is NOT a verb but an adjective or noun.
- Page 11 Drop the use of "any'' in the sentence "we will suppose without any loss of generality that...'' (it's redundant).
- Page 13 Problems involving the Stiefel Manifold can often be solved via Riemannian optimization methods; consider trying https://www.manopt.org/, see https://www.nicolasboumal.net/book/IntroOptimManifolds_Boumal_2022.pdf for a reference.
- Page 14 What is the function $\kappa$ in the definition of $P^{\{1\}}$?
- Page 15 Use \colon (the command) as supposed to ":'' when defining functions.
- Page 17 Figure 5 is impossible to parse. Are you somehow comparing the error between the two methods? It seems that you are just plotting labels.
- For the experiments, use a color-blind-friendly palette.

**Strengths And Weaknesses:**

*Strengths*
  - The proposed distance is novel.
  - Further, in practice, the proposed methods appear to be on par or better than existing methods in terms of time and performance.

*Weaknesses*
  - This distance is limited to GMMs. To compute the distance for discrete samples, one needs to fit a GMM to the data. In turn, this yields an additional source of error that the current paper does not consider.
  - Further, fitting the GMM turns out to be theoretically delicate, as it has been shown that the optimization landscape of maximum likelihood is not nicely behaved, and the EM algorithm can get stuck at spurious critical points.

---

### Review · Reviewer_oCzP · 2024-07-26

**Summary Of Contributions:**

This paper introduces two new shape/embedded-distribution matching formulations, and algorithms.  The idea is to leverage the Gromov-Wasserstein framework which finds the optimal transportation plan between the pairwise distances among points within each input.  The Wasserstein part is known to be slow, and the Gromov aspect even slower; so this paper explores first approximating the distributions with a Gaussian Mixture Model of size at most K.  They introduce two specific variants:
  - MGW is as described above and there is a closed form way to compute the distance that depends only on K.  The authors show it is a psuedometric, and through a few examples shows it approximates the GW pretty well.
  - MEW replaces the Gromov aspect (which learns over all pairwise distances) with "embedded" aspect that tries to learn the best affine-transformation between compared shapes.  The optimization is non-convex, and the paper offers an annealing approach that seems to work reasonably well.  As a benefit it produces a transportation plan.

A point of comparison is the quantized Wasserstein that use a Voronoi quantization into K points instead of a Gaussian Mixture Model.  This is also known to converge to the full Wasserstein in the limit, and seems to usually do better in their experiments.

**Audience:**

Yes

**Claims And Evidence:**

Yes

**Requested Changes:**

- More clearly state the advantages of MGW (and MEW) over qGW.
  - The algorithm labels in the legends of the figures did not quite match the description in the text.  Please fix.

**Strengths And Weaknesses:**

Strengths:
 - The paper is written clearly, and with both mathematical rigor and clarify.  The new approaches are well described, and I found accessible despite the formality.
 - Combining GMM with Wasserstein and Gromov-Wasserstein is a very natural approach, and so this study is well-warranted.
 - The empirical study seemed fine, and the benefits of methods were made clear and convincing.

Weaknesses:
 - The MEW algorithm is not guaranteed to converge to optimum.  Some examples demonstrate the issues.  This is an instance of a known challenging task in this space of shape matching.
 - The MGW seems to do well, but not as well as the quantized-GW approach in terms of time/accuracy trade-off.

---

> ### Author Response · Authors · 2024-08-07
> **Response to Reviewer oCzP**
>
> Thank you for your review and your comments. About the comparison between $ MGW_2 $ and qGW, first note that the $ MGW_2 $ method primarly provides a distance between clouds of points, before providing an heuristic coupling between them. In contrast, the qGW method directly derives an heuristic coupling which can then be reinjected in the objective function to derive a distance. For both methods, the optional additional step (computing a coupling for $ MGW_2 $ vs computing a distance for qGW) increases the computation time. Consequently, $ MGW_2 $ seems to be a more adequate choice than qGW for tasks that only require to assess a distance between clouds of points (see the computation times of Figure 7), whereas qGW seems a more adequate choice for tasks that require only a coupling between points.  Still, we have shown in our experiments that $ MGW_2 $ can also provide couplings, with equivalent performance to qGW in terms of accuracy despite being indeed significantly slower in the small-to-medium scale setting of Table 1. Yet, we believe that an advantage of $ MGW_2 $ over qGW for deriving couplings is that $ MGW_2 $ seems to better decorrelate the number of clusters required to achieve good accuracy from the number of points, which might become an important feature in larger scale settings.
> This discussion has been added in the conclusion of the paper.
>
> ----
>
>  > The algorithm labels in the legends of the figures did not quite match the description in the text. Please fix.
>
> Thank you for pointing this out, this has been fixed.

---

> > ### Comment · Reviewer_oCzP · 2024-08-12
> >
> > Thanks for the update.  I think this better places the nice new results in relative context.  I am satisfied with the paper now.

---

### Decision · Action_Editor_V97o · 2024-08-31

**Recommendation:** Accept as is

**Comment:**

The main topic is a clear interest:  can Wasserstein computation be improved by first approximating with Gaussian Mixture Modeling.  The answer is mathematically involved, but ultimately the improvements are incredibly modest.  It is worth having this worked out.

**Audience:**

The general topic is one of definite interest for the ML community.  The reviewers all agree.

**Claims And Evidence:**

All of the reviews agree that the paper supports the claims made in the paper, mathematically and empirically.